# Calcium-mediated shaping of naive CD4 T-cell phenotype and function

Vincent Guichard[1,2], Nelly Bonilla[1], Aurélie Durand[1], Alexandra Audemard-Verger[1], Thomas Guilbert[1], Bruno Martin[1], Bruno Lucas[1†], Cédric Auffray[1†]*

[1]Institut Cochin, Paris Descartes Université, CNRS UMR8104, INSERM U1016, Paris, France; [2]Paris Diderot Université, Paris, France

**Abstract** Continuous contact with self-major histocompatibility complex ligands is essential for the survival of naive CD4 T cells. We have previously shown that the resulting tonic TCR signaling also influences their fate upon activation by increasing their ability to differentiate into induced/peripheral regulatory T cells. To decipher the molecular mechanisms governing this process, we here focus on the TCR signaling cascade and demonstrate that a rise in intracellular calcium levels is sufficient to modulate the phenotype of mouse naive CD4 T cells and to increase their sensitivity to regulatory T-cell polarization signals, both processes relying on calcineurin activation. Accordingly, in vivo calcineurin inhibition leads the most self-reactive naive CD4 T cells to adopt the phenotype of their less self-reactive cell-counterparts. Collectively, our findings demonstrate that calcium-mediated activation of the calcineurin pathway acts as a rheostat to shape both the phenotype and effector potential of naive CD4 T cells in the steady-state.
DOI: https://doi.org/10.7554/eLife.27215.001

*For correspondence:
cedric.auffray@inserm.fr

†These authors contributed equally to this work

**Competing interests:** The authors declare that no competing interests exist.

## Introduction

T-cell precursors originate in the bone-marrow and are educated in the thymus through processes called positive and negative selections, which result in MHC-restriction and self-tolerance, respectively (*Stritesky et al., 2012*). Only those T cells that bear an αβT-cell receptor (TCR) recognizing self-MHC with a relatively low affinity will differentiate and exit into the systemic circulation as self-MHC restricted T cells. T cells carrying an αβ TCR that reacts with self-MHC with very low affinity die by neglect, whereas those recognizing self-MHC with high affinity are mostly deleted by apoptosis or differentiate into regulatory T cells called 'natural' or thymically derived (tTreg) in order to prevent autoimmunity (*Bautista et al., 2009*; *Leung et al., 2009*). Therefore, self-MHC and the associated self-reactivity of T cells influence both T-cell production and phenotype in the thymus.

In the periphery, the pre-immune repertoire of T cells is composed of almost 70% of naive T cells. The remaining 30% are divided between recent thymic emigrants with a comparable phenotype, regulatory T cells (Foxp3+) and cells with an activated/memory phenotype. Naive T cells are kept alive through continuous TCR interactions with MHC molecules complexed with various self-peptides. Such TCR/MHC interactions plus contacts with IL-7 cause low-level signaling, which promotes long-term survival of naive T cells in interphase through the synthesis of anti-apoptotic molecules such as Bcl-2 (*Martin et al., 2006*; *Takada and Jameson, 2009*).

The degree of TCR self-reactivity of a given T-cell clone has been correlated with its expression of CD5 and Nur77 (*Azzam et al., 1998*; *Moran et al., 2011*). We have recently identified the cell surface GPI-anchored protein, Ly-6C, as an additional and complementary sensor of T-cell self-reactivity (*Martin et al., 2013*). Indeed, we have shown that, in contrast to CD5 and Nur77 which expression directly correlates with self-reactivity, the expression of Ly-6C by peripheral naive CD4 T cells (CD4

**eLife digest** To help protect the body from disease, small immune cells called T lymphocytes move rapidly, searching for signs of infection. These signs are antigens – processed pieces of proteins from invading microbes – that are displayed on the surface of so-called antigen-presenting cells. Before it encounters its specific antigen, a T cell is called naive. After encountering its antigen, the naive T cell activates and then develops into a variety of immune cells, each with a specific activity. These immune cells include so-called peripherally induced regulatory T cells (or "pTreg cells" for short), which, as the name suggests, help to regulate the immune response.

In addition to foreign antigens from microbes, antigen-presenting cells display fragments of the body's own proteins too. All naive T cells recognize some "Self-antigens", but not as strongly as they recognize foreign antigens. As a naive T cell travels around the body, it repeatedly interacts with antigen-presenting cells that display Self-antigens, which triggers a low level of signaling in the T cell. While this background signaling was known to help the T cell survive, in 2013, researchers reported that: it also makes the T cell more responsive to foreign antigens; and it shapes how these cells will respond when activated. For example, the naive T cells that respond the most to Self-antigens were seen to be much more likely to become pTreg cells when activated than other T cells.

Guichard et al. – who include several of the researchers involved in the 2013 work – set out to understand why the most Self-reactive T cells show this bias toward becoming pTreg cells. The experiments used a range of approaches with T cells both in the laboratory and in mice. By looking at which genes were active in the most Self-reactive T cells, Guichard et al. narrowed in on a signaling pathway that involves calcium ions and an enzyme called Calcineurin. Blocking this pathway caused the most Self-reactive T cells to lose their bias, and instead develop in the same way as the least Self-reactive T cells.

Guichard et al. propose that the continuous interactions with Self-antigens trigger waves of calcium ions in a naive T cell that shapes its behavior and future development. In a related study, Dong, Othy et al. also conclude that contact with antigen-presenting cells causes calcium signals that shape how the T cells behave.

In addition to providing more detail about the inner workings of immune cells, these findings may also have implications in a clinical setting. Calcineurin inhibitors are often used to suppress the immune system in transplant patients to prevent rejection of the transplanted organ. However, it has proved difficult to safely interrupt these therapies even after many years. These new findings may provide a possible explanation for this, by suggesting that the inhibitors may also interfere with the generation of pTreg cells. Without these cells' regulatory influence, the immune system is unlikely to ever become tolerant of the transplant.

DOI: https://doi.org/10.7554/eLife.27215.002

$T_N$ cells) inversely correlates with their ability to interact with self-MHC (*Martin et al., 2013*). Ly-6C⁻ CD4 $T_N$ cells were therefore identified as more self-reactive than their Ly-6C⁺-cell counterparts.

In the absence of foreign antigen, peripheral naive T cells continuously recirculate between lymphoid organs (*Gowans, 1959*), in which they migrate along the fibroblastic reticular cells network (*Bajénoff et al., 2006*) and interact frequently and briefly with dendritic cells (DC) (*Bajénoff et al., 2006*; *Mempel et al., 2004*). It is generally accepted that these frequent DC-T-cell interactions increase the probability of contacts between very rare antigen-specific naive T cells and the few DCs presenting their cognate antigen during the initial course of an infection. Experimental evidences indicate that self-MHC recognition in the periphery is also required to maintain T cells in a state of responsiveness toward foreign antigen (*Persaud et al., 2014*; *Stefanová et al., 2002*; *Wülfing et al., 2002*), suggesting a crucial role for self-MHC mediated 'education' and TCR self-reactivity in determining the intrinsic functional attributes of CD4 $T_N$ cells. Altogether, this steady-state tonic TCR signaling was shown to influence CD4 $T_N$-cell effector fate by increasing the magnitude of their response toward their cognate antigens.

Following activation by antigen-presenting cells (APCs) in the periphery, the bulk of CD4 $T_N$ cells can differentiate into a variety of well documented T-helper ($T_H$) cell subsets, such as $T_H1$, $T_H2$, $T_H17$ or peripherally induced regulatory T cells (pTreg cells), characterized by their cytokine production

profiles, specific effector functions and lineage-specific transcription factors (T-bet for $T_H1$ cells, GATA-3 for $T_H2$ cells, RORγt/RORα for $T_H17$ cells and Foxp3 for pTreg cells) (*Abbas et al., 1996*; *Bilate and Lafaille, 2012*; *Fontenot et al., 2003*; *Hori et al., 2003*; *Ivanov et al., 2006*; *Liang et al., 2006*; *Mosmann et al., 1986*; *Szabo et al., 2000*; *Ye et al., 2001*; *Zheng and Flavell, 1997*). Among these effector CD4 T cells, pTreg cells produce TGF-β and share phenotypic and functional characteristics with tTreg cells (*Bilate and Lafaille, 2012*). The immunological context in which CD4 $T_N$ cells are immersed at the time of their activation is known to drive lineage commitment. The strength of the activating TCR signals received by a CD4 $T_N$ cell also influences its subsequent polarization toward particular differentiation pathways (*Corse et al., 2011*). Indeed, in weakly polarizing conditions, low TCR signals favor $T_H2$- and pTreg-cell differentiation, whereas $T_H1$- and $T_{FH}$-cell differentiation arises from stronger signals (*Gottschalk et al., 2010*; *Rogers and Croft, 1999*; *Turner et al., 2009*). Most of these data were obtained in vitro by modulating signal strength with graded dose of peptide-MHC ligands of varying potency. However, only relatively high-affinity TCR–MHC interactions were shown to facilitate the induction of persistent Foxp3$^+$ T cells in vivo (*Gottschalk et al., 2010*).

Our recent work has reinforced the link between the tonic TCR signaling received by CD4 $T_N$ cells in the steady-state and their fate in the effector phase. Indeed, we have demonstrated that TCR/self-MHC interactions not only increase quantitatively but also shape qualitatively the response of CD4 $T_N$ cells to their cognate antigens in the effector phase (*Martin et al., 2013*). More precisely, by taking advantage of our data showing that Ly-6C expression can be considered as a new sensor of CD4 $T_N$-cell self-reactivity, we have demonstrated that CD4 $T_N$ cells with the highest avidity for self-MHC (Ly-6C$^-$) have a biased commitment toward the iTreg/pTreg-cell lineage (*Martin et al., 2013*).

The binding of antigen/MHC complexes to the TCR triggers the recruitment of a series of signaling molecules and adaptors to the TCR/CD3 complex that ultimately results in the phosphorylation and activation of phospholipase C-γ (PLCγ). PLCγ then cleaves the phospholipid phosphatidylinositol 4,5-bisphosphate (PIP$_2$) in the plasma membrane to generate diacylglycerol, which activates protein kinase C (PKC) and Ras-dependent pathways, as well as 1,4,5-inositol trisphosphate (IP$_3$), which induces the release of calcium (Ca$^{2+}$) from intracellular stores (the endoplasmic reticulum (ER)). This Ca$^{2+}$ store release only transiently elevates intracellular Ca$^{2+}$ concentrations but this transient rise induces in turn a massive and sustained Ca$^{2+}$ entry from the extracellular space (*Hogan et al., 2010*).

With the aim of deciphering the molecular mechanisms involved in the tonic TCR-signaling-mediated shaping of the CD4 $T_N$-cell compartment, we have focused on the TCR signaling cascade. By using transcriptomic and phenotypic approaches as well as in vitro and in vivo assays, we have identified the Ca$^{2+}$ signaling pathway as key for the acquisition of both the phenotype of the most self-reactive CD4 $T_N$ cells and their enhanced cell-intrinsic ability to commit into regulatory T cells upon activation in vitro (iTreg) and in vivo (pTreg).

## Results

### Cell-intrinsic enhanced ability of Ly-6C$^-$ CD4 $T_N$ cells to commit into iTreg cells

We have recently shown that CD4 $T_N$ cells with the highest avidity for self-MHC (Ly-6C$^-$ CD4 $T_N$ cells) have a biased commitment toward the iTreg/pTreg-cell lineage (*Martin et al., 2013*). As $T_H1$- and $T_H2$-cell-derived cytokines are known to inhibit iTreg-cell induction in vitro (*Henderson et al., 2015*), we first wondered whether Ly-6C$^-$ and Ly-6C$^+$ CD4 $T_N$ cells had the same ability to produce such cytokines after stimulation. Ly-6C$^-$ and Ly-6C$^+$ CD4 $T_N$ cells were thus stimulated with αCD3- and αCD28-coated antibodies in the presence or absence of TGFβ. Interferon-gamma (IFN-γ) and interleukins (IL) -4, -17 and -10 were assayed in the supernatants collected 24 hr after the beginning of the culture. We found that, whatever the presence or absence of TGFβ in the culture medium, Ly-6C$^-$ and Ly-6C$^+$ CD4 $T_N$ cells produced similar amounts of these cytokines (*Figure 1—figure supplement 1A,B*). To further characterize the enhanced ability of Ly-6C$^-$ CD4 $T_N$ cells to commit into iTregs in vitro, we asked whether this feature was cell-intrinsic. To this end, Ly-6C$^-$ and Ly-6C$^+$ CD4 $T_N$ cells were isolated from peripheral LNs of C57BL/6 Foxp3-GFP mice by flow cytometry sorting, barcoded with CTv or CTv and CTfr proliferation dyes, and stimulated with αCD3- and αCD28-

coated antibodies in the presence of graded doses of TGFβ. These cells were cultured separately or together (*Figure 1A,B*). The percentages of Foxp3$^+$ cells among the progeny of both naive cell-subsets were assessed on day 4. For suboptimal doses of exogenous TGFβ,Ly-6C$^-$ CD4 T$_N$ cells gave rise to a twofold higher proportion of iTreg cells than their Ly-6C$^+$-cell counterparts in both culture conditions (*Figure 1C,D*). The concentration of TGFβ needed to obtain 50% of the maximal percentage of iTreg cells (effective concentration, EC50) was calculated by fitting the dose-response curves of both CD4 T$_N$-cell subsets in the different culture conditions (*Figure 1D,E*). EC50 values for TGFβ were statistically different between the 2 CD4 T$_N$-cell subsets whether they were cultured separately or together. Of note, and in line with their similar ability to produce T$_H$1- and T$_H$2-cell-derived cytokines, blocking IFN-γ and IL-4 during in vitro iTreg-cell polarization did not abolish the difference in the ability of Ly-6C$^-$ and Ly-6C$^+$ CD4 T$_N$ cells to differentiate into iTreg cells. (*Figure 1—figure supplement 1C–E*). These results suggest strongly that the greater sensibility of Ly-6C$^-$ CD4 T$_N$ cells to iTreg-cell polarization signals is cell-intrinsic.

## Ly-6C$^-$ CD4 T$_N$-cell transcriptomic signature reveals both their TCR signaling activity and their bias toward iTreg-cell polarization

To further compare Ly-6C$^-$ and Ly-6C$^+$ CD4 T$_N$ cells, we obtained Affymetrix gene expression profiles from both CD4 T$_N$-cell subsets directly isolated from peripheral LNs of C57BL/6 Foxp3-GFP mice by flow cytometry sorting (*Figure 2*). Only few genes were significantly differentially expressed between the two types of CD4 T$_N$ cells (at a 1.3-fold cutoff, 167 genes over-expressed and 164 under-expressed in Ly-6C$^-$ CD4 T$_N$ cells when compared to Ly-6C$^+$ CD4 T$_N$ cells; *Figure 2A*). This set of differentially expressed genes between Ly-6C$^-$ and Ly-6C$^+$ CD4 T$_N$ cells was compiled into a comprehensive signature that we named 6CSign (*Figure 2B,C*). The differential expression of several genes by Ly-6C$^-$ and Ly-6C$^+$ CD4 T$_N$ cells was then validated at the protein level by flow-cytometry (*Figure 2—figure supplement 1*). In line with our microarray analysis, Ly-6C$^-$ CD4 T$_N$ cells were expressing higher amounts of CD5, CD73, CD122, CD200, Ikzf3 and Izumo1r and lower levels of Sca-1 and IL18Rα than their Ly-6C$^+$ CD4 T$_N$-cell counterparts (*Figure 2—figure supplement 1A,B*).

We have previously shown that Ly-6C$^-$ CD4 T$_N$ cells were more self-reactive than Ly-6C$^+$ CD4 T$_N$ cells (*Martin et al., 2013*). Accordingly, among the 6CSign, several genes such as *Ctla4*, *Cd5*, *Tnfrsf4*, *Tnfrsf9* and *Nr4a1* were previously shown to belong to activation-induced or -repressed gene families (*Figure 2C*; [*Wakamatsu et al., 2013*]). We thus compared more precisely our signature, the 6CSign, with several public Geo Datasets comparing various 'activated' CD4 T$_N$ cells to their non-activated cell counterparts (*Figure 3A,B*). CD5 expression levels on CD4 T$_N$ cells are actively maintained by interactions with self-MHC and rapidly decline in their absence (for example in the blood, [*Stefanová et al., 2002*]). In agreement with a greater self-reactivity of Ly-6C$^-$ CD4 T$_N$ cells, the 6CSign correlated significantly with the CD5$^{hi}$ versus CD5$^{lo}$ CD4 T$_N$-cell signature (*Richards et al., 2015*). Interestingly, whereas the 6CSign genes also correlated with the transcriptional signature of αCD3-activated CD4 T$_N$ cells (compared to unstimulated cells) (*Wakamatsu et al., 2013*), there was no significant correlation with the signature of Phorbol 12-Myristate 13-Acetate (PMA)-activated CD4 T$_N$ cells (*Bevington et al., 2016*).

Interestingly, the 6CSign contained several genes characteristically expressed in Treg cells such as *Ctla4*, *Izumo1r*, *Cd200*, *Lag3* or *Il2rb*. All these genes were upregulated in Ly-6C$^-$ CD4 T$_N$ cells when compared to Ly-6C$^+$ CD4 T$_N$ cells (*Figure 2C*). By comparing CD4 T-cell effectors with naive CD4 T cells, Wei et al. (2009) have recently defined the transcriptional signature of the main CD4 T$_H$-cell subsets such as in-vitro-induced Treg cells, T$_H$1, T$_H$2 and T$_H$17 cells. Comparison of the 6CSign with these cell signatures revealed that the differences in gene expression observed between Ly-6C$^-$ and Ly-6C$^+$ CD4 T$_N$ cells correlated significantly with the in-vitro-induced Treg-cell signature, and to a lesser extent with the T$_H$17 one but not with the T$_H$1 or T$_H$2 transcriptional signatures (*Figure 3C*).

Similarities were also observed between the transcriptional profiles of Ly-6C$^-$ CD4 T$_N$ cells and ex vivo purified peripheral Treg cells (*Figure 3C*) (*Wei et al., 2009*). One common characteristic shared by Ly-6C$^-$ CD4 T$_N$ cells and CD4 Treg cells is their high degree of self-reactivity. Recent studies have highlighted a continuous requirement of self-MHC recognition and of the associated TCR-mediated signaling for maintaining both the function and transcriptional signature of CD4 Treg cells (*Delpoux et al., 2012*; *Levine et al., 2014*; *Vahl et al., 2014*). Whereas self-deprivation or TCR-ablation did not impair the expression of the transcription factor Foxp3, they induced major transcriptional changes (*Delpoux et al., 2012*; *Levine et al., 2014*; *Vahl et al., 2014*). Interestingly, 6CSign

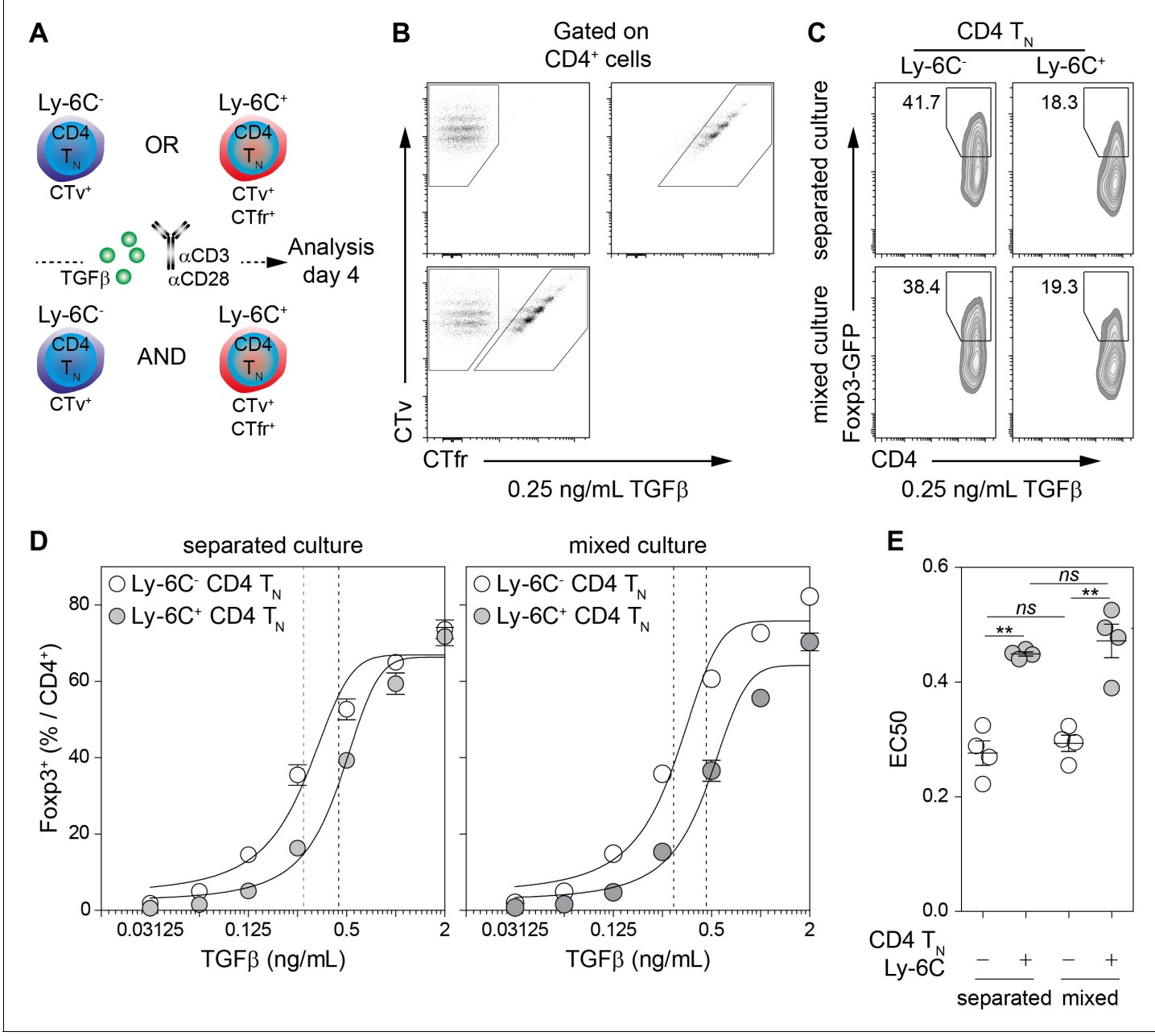

**Figure 1.** Cell-intrinsic enhanced ability of Ly-6C⁻ CD4 T_N cells to commit into iTreg cells. (A–E) Flow-cytometry sorted Ly-6C⁻ and Ly-6C⁺ CD4 T_N cells from C57BL/6 Foxp3-GFP mice were stained with CTv (Ly-6C⁻) or CTv and CTfr (Ly-6C⁺) and stimulated separately or together for 4 days with coated αCD3 and αCD28 Abs (4 μg/mL), in the presence of graded doses of TGFβ1. (A) Diagram illustrating the experimental protocol. (B) Representative CTv/CTfr dot-plots for gated CD4⁺ cells recovered after 4 days of culture. Ly-6C⁻ and Ly-6C⁺ CD4 T_N cells were either cultured separately (top left and right panels, respectively) or together (bottom panel) (C) Representative Foxp3/CD4 contour-plots and proportions of Foxp3⁺ cells for gated CD4⁺ cells are shown at a dose of 0.25 ng/mL TGFβ1. (D) Proportions of Foxp3⁺ cells among CD4⁺ cells are shown as a function of TGFβ1 concentration. Mean ± s.e.m of four independent experiments are shown. (E) Concentrations of TGFβ1 needed to obtain 50% of the maximal percentages of iTreg-cell polarization (EC50) were calculated for each CD4 T_N cell subset in separated or mixed cultures. Each dot represents an independent experiment. Significance of differences were assessed using a two-tailed paired Student's t-test. Values of p<0.05 were considered as statistically significant (**p<0.01; *ns*, not significant).

DOI: https://doi.org/10.7554/eLife.27215.003

The following figure supplement is available for figure 1:

**Figure supplement 1.** Cytokine-independent enhanced ability of Ly-6C⁻ CD4 T_N cells to commit into iTreg cells.
DOI: https://doi.org/10.7554/eLife.27215.004

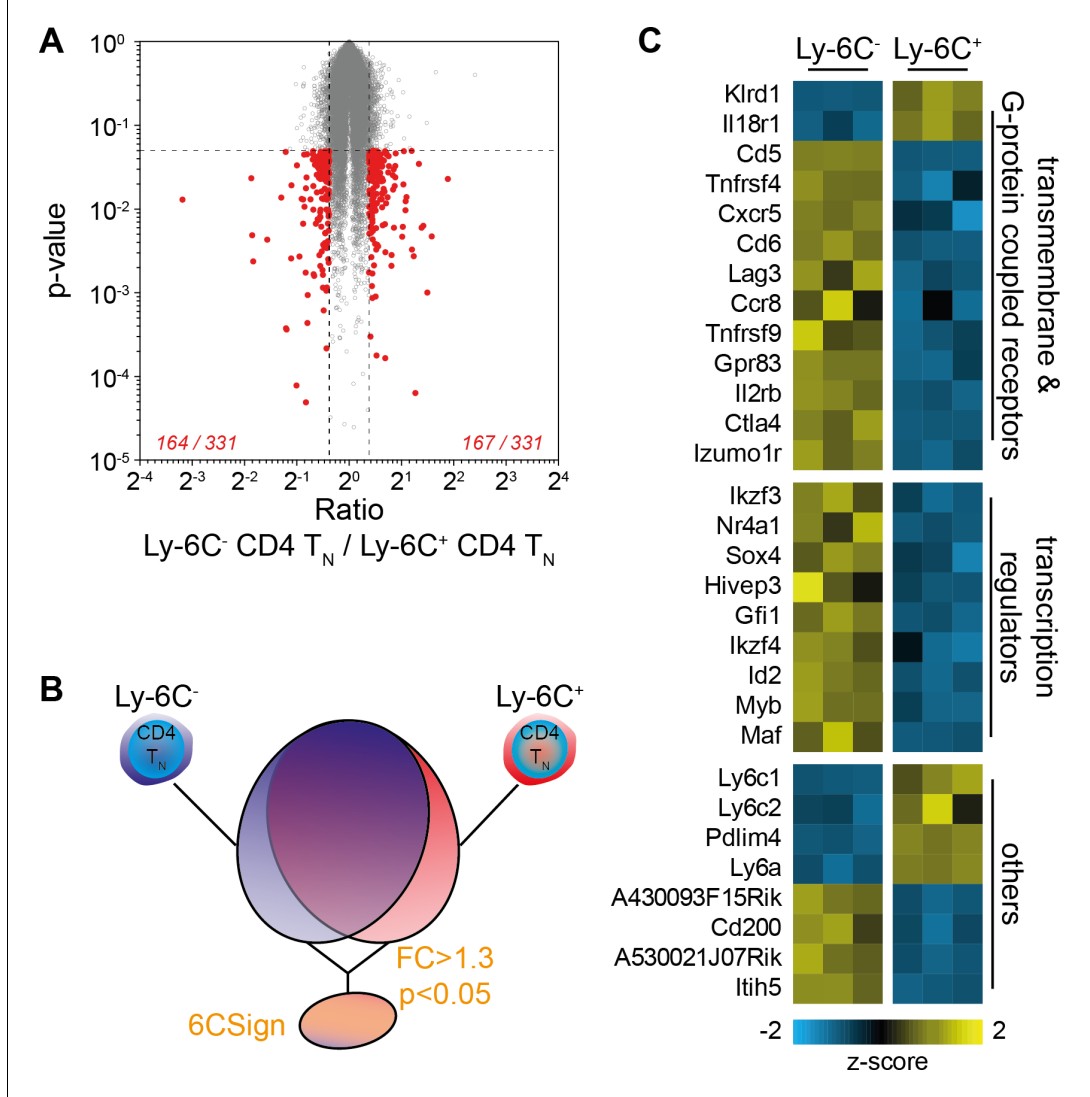

**Figure 2.** Transcriptional profiling identifies a set of differentially regulated genes between Ly-6C⁻ and Ly-6C⁺ CD4 T_N-cell subsets. (A–E) Microarray analysis was performed on Ly-6C⁻ and Ly-6C⁺ CD4 T_N cells sorted from LNs of C57BL/6 Foxp3-GFP mice. (A) 'Volcano plot' representation (Log2 (ratio) versus Log10 (t test p-value)). Genes expressed >1.3 fold higher or lower in Ly-6C⁻ CD4 T_N cells compared to Ly-6C⁺ CD4 T_N cells with a p-value of <0.05 are highlighted in red. The number of genes up- or down-regulated (1.3-fold cut-off) for each comparison is indicated. (B) Scheme depicting the selection of genes that were included in the 6CSign (list of the genes differentially expressed between Ly-6C⁻ and Ly-6C⁺ CD4 T_N cells, at a 1.3-fold cut-off). (C) Heat map of selected differentially expressed genes between Ly-6C⁻ and Ly-6C⁺ CD4 T_N cells. The scaled expression of each replicate, denoted as the row z-score, is plotted in yellow-blue color scale with yellow indicating high expression and blue indicating low expression.

DOI: https://doi.org/10.7554/eLife.27215.005

The following figure supplement is available for figure 2:

**Figure supplement 1.** Validation of the 6CSign at the protein level.

DOI: https://doi.org/10.7554/eLife.27215.006

genes strongly correlated with the transcriptional signature of TCR⁺ CD4 Treg cells (compared to TCR⁻ CD4 Treg cells, *Figure 3D*) (*Vahl et al., 2014*). More precisely, most genes upregulated in Ly-6C⁻ CD4 T_N cells when compared to their Ly-6C⁺-cell counterparts were positively regulated by steady-state TCR signaling in CD4 Treg cells (such as *Cd5*, *Cd200*, *Il2rb*, *Itih5*, *Maf* and *Myb*; *Figure 3E*). Conversely, an important proportion of the genes downregulated in Ly-6C⁻ CD4 T_N cells were also down-regulated by steady-state interactions with self-MHC in CD4 Treg cells (*Figure 3E*). Altogether, these data point to a role for the TCR signaling pathway in the installation and mainte-nance of the 6CSign.

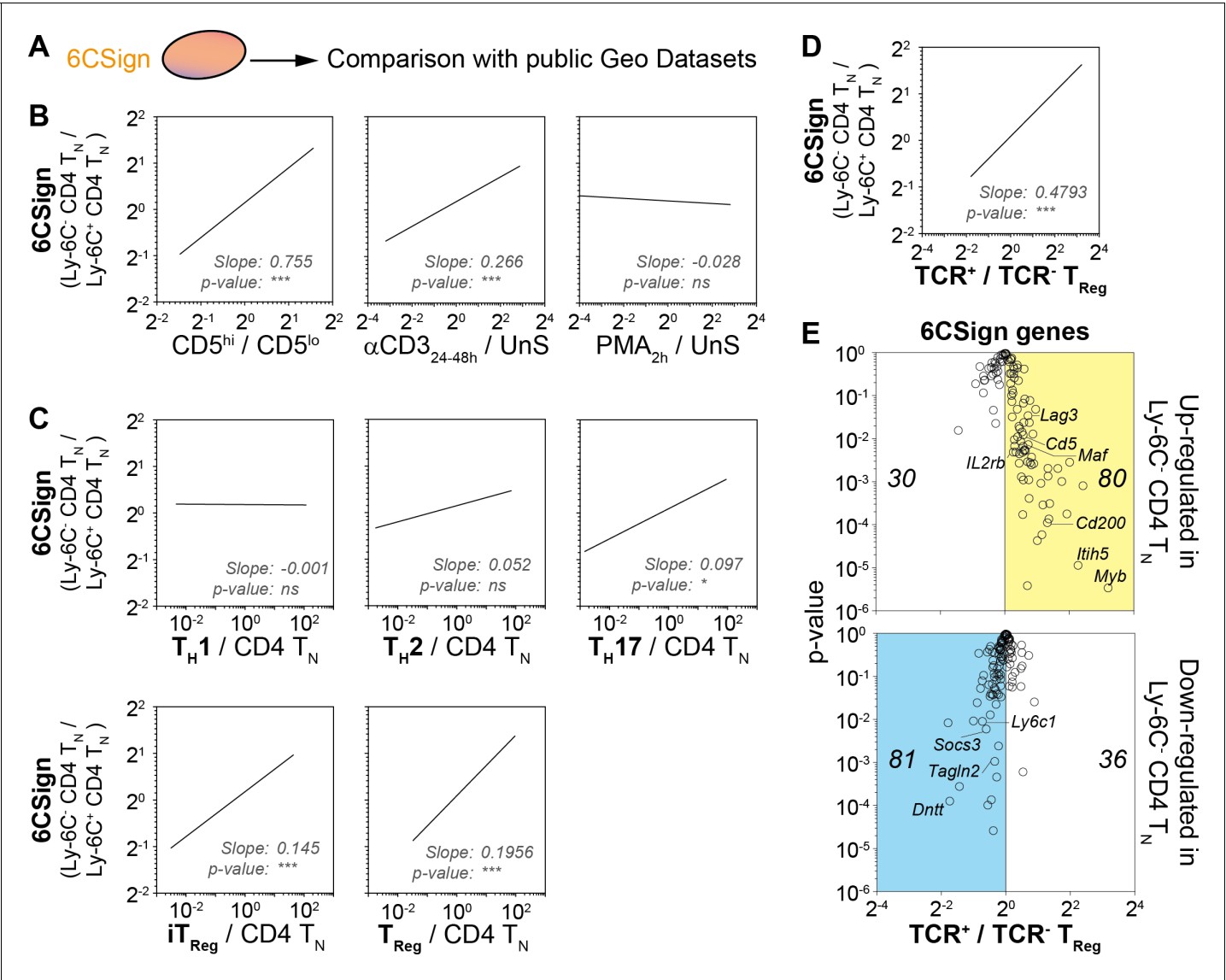

**Figure 3.** Transcriptomic signature of Ly-6C⁻ CD4 $T_N$ cells reveals both their active TCR signaling and their bias toward iTreg-cell polarization. (A–E) 6CSign was compared to several public Geo Datasets. (A) Diagram illustrating the analysis protocol. (B) Ratio vs ratio representation comparing gene expression ratio between Ly-6C⁻ CD4 $T_N$ cells and Ly-6C⁺ CD4 $T_N$ cells (6CSign) with either CD5$^{hivslo}$ cell signature (*Richards et al., 2015*) (ratio of CD5$^{hi}$ CD4 $T_N$ cells to CD5$^{lo}$ CD4 $T_N$ cells, left panel), anti-CD3 activated CD4 $T_N$-cell signature (*Wakamatsu et al., 2013*) (ratio of CD4 $T_N$ cells stimulated for 24–48 hr with anti-CD3 coated Ab to unstimulated CD4 $T_N$ cells, middle panel) and PMA-activated CD4 $T_N$-cell signature (*Bevington et al., 2016*) (ratio of CD4 $T_N$ cells stimulated for 2 hr with PMA to unstimulated CD4 $T_N$ cells, right panel). (C) Ratio vs ratio representation comparing gene expression ratio between Ly-6C⁻ CD4 $T_N$ cells and Ly-6C⁺ CD4 $T_N$ cells (6CSign) with in-vitro-induced $T_H1$, $T_H2$, $T_H17$, iTreg and ex vivo purified Treg cell signatures that have been identified by *Wei et al. (2009)* (ratio of CD4 $T_H$-cell subsets to CD4 $T_N$ cells). (D) Ratio vs ratio representation comparing gene expression ratio between Ly-6C⁻ CD4 $T_N$ cells and Ly-6C⁺ CD4 $T_N$ cells (6CSign) with TCR-signaling-dependent CD4-Treg-specific signature (*Vahl et al., 2014*) (ratio between TCR⁺ Treg cells and TCR-ablated (TCR⁻) Treg cells). (B–D) Correlation analyses were performed using Pearson's correlation test. (E) 'Volcano plot' representation (Log₂ (ratio) versus Log₁₀ (t-test p-value)) between TCR⁺ Treg cells and TCR-ablated (TCR⁻) Treg cells (*Vahl et al., 2014*), for 6CSign genes upregulated (upper panel) or downregulated (lower panel) in Ly-6C⁻ CD4 $T_N$ cells. (B–E) Datasets were filtered to common probes between the two arrays.

DOI: https://doi.org/10.7554/eLife.27215.007

## Ly-6C⁻ CD4 $T_N$-cell phenotype relies on the Ca²⁺ signaling pathway in vitro

The transcriptional signature of Ly-6C⁻ CD4 $T_N$ cells revealed some similarities between these cells and αCD3-stimulated CD4 $T_N$ cells. We therefore decided to analyze the effect of TCR signaling on

Ly-6C expression. Ly-6C$^+$ CD4 T$_N$ cells were isolated from peripheral LNs of C57BL/6 Foxp3-GFP mice by flow cytometry sorting and incubated with various stimulating agents mimicking all or part of TCR-induced signals (*Figure 4A*). As expected from our transcriptomic analysis and previous work (*Martin et al., 2013*), Ly-6C expression was clearly downregulated when cells were stimulated with αCD3 and αCD28-coated antibodies for 5 days (*Figure 4A,B*). To decipher which TCR-induced signals led to Ly-6C down-regulation, we roughly dichotomized the TCR signaling cascade into its two main components, for example the Ca$^{2+}$ signaling pathway that can be elicited by Thapsigargin (TG) and the PKC and ERK signaling pathways activated by PMA. When combined, PMA and TG, induced complete Ly-6C down-regulation, whereas, when separated, each drug had an opposite effect on Ly-6C expression. Indeed, whereas PMA alone upregulated Ly-6C expression, TG alone induced a near-complete disappearance of Ly-6C protein at the surface of Ly-6C$^+$ CD4 T$_N$ cells. Interestingly, while in all other conditions, CD4 T$_N$ cells were proliferating, this phenotypic conversion of Ly-6C$^+$ CD4 T$_N$ cells into Ly-6C$^-$ CD4 T$_N$ cells induced by TG alone occurred without any proliferation (*Figure 4A,B*). Importantly, to avoid TG-induced cell death, a sub-optimal dose (4 nM) was used in these culture conditions. 4 nM TG induced a reproducible increase in intracellular calcium levels, although to a lesser extent than the classical dose of 200 nM (*Figure 4C*). Accordingly, by analyzing basal Ca$^{2+}$ contents at the end of the culture period (5 days), we observed that 4 nM TG treated Ly-6C$^+$ CD4 T$_N$ cells exhibited higher cytoplasmic Ca$^{2+}$ levels than control cells cultured in IL-7 alone (*Figure 4D,E*). To further characterize the long-term effect of this low-dose TG, subcellular localization of the nuclear factor of activated T-cell protein 1 (NFAT1) was assessed in Ly-6C$^+$ CD4 T$_N$ cells in the presence or absence of 4 nM TG at various time points along the culture. Indeed, increases in intracellular Ca$^{2+}$ levels result in the activation of calcineurin that dephosphorylates members of the NFAT family, leading to their translocation into the nucleus. NFAT1 localization was quantified by high-resolution imaging flow-cytometry using the ImageStreamX technology (*Figure 4F*). In line with the Ca$^{2+}$ increase induced by 4 nM TG treatment, NFAT was translocated into the nucleus of Ly-6C$^+$ CD4 T$_N$ cells in the presence of TG while it remained cytoplasmic in its absence. NFAT translocation into the nucleus peaked on day 1 and remained significantly higher in TG-treated cells than in control cells throughout the culture.

Finally, in agreement with their resting status, Ly-6C$^+$ CD4 T$_N$ cells cultured in TG alone for 5 days maintained a naive phenotype according to their low forward scatter profile and expression of CD44 and CD62L (*Figure 4—figure supplement 1*). We then studied the kinetic aspect of the TG-mediated conversion of Ly-6C$^+$ CD4 T$_N$ cells into Ly-6C$^-$ CD4 T$_N$ cells and found that it occurred in 3–4 days of culture (*Figure 4G*).

Altogether, our results suggest that the Ca$^{2+}$ signaling pathway is sufficient to induce Ly-6C down-regulation on CD4 T$_N$ cells in vitro. We therefore hypothesized that the Ca$^{2+}$ signaling pathway might be involved as part of the self-mediated tonic TCR signaling in the generation/maintenance of Ly-6C$^-$ CD4 T$_N$ cells in the periphery of a normal mouse in the steady-state. To evaluate the activation status of the Ca$^{2+}$ signaling pathway within Ly-6C$^-$ and Ly-6C$^+$ CD4 T$_N$ cells in vivo in the steady-state, we analyzed NFAT1 subcellular localization in both cell types. To this aim, LN cells were directly fixed after recovery and NFAT1 or NFAT2 localization was imaged by confocal microscopy (*Figure 4H*) and quantified by high-resolution imaging flow-cytometry using the ImageStreamX technology (*Figure 4I* and *Figure 4—figure supplement 2*). In line with our hypothesis, NFAT localization is more nuclear in Ly-6C$^-$ CD4 T$_N$ cells than in their Ly-6C$^+$-cell counterparts. As a control, CD4 T$_N$ cells were rested in vitro for 30 min before fixation and staining. As expected, differences in NFAT localization between Ly-6C$^-$ and Ly-6C$^+$ CD4 T$_N$ cells were abolished in these conditions. 200 nM TG treatment induced the nuclear translocation of NFAT in both CD4 T$_N$-cell subsets. Of note, in this latter condition, differences in the localization of NFAT1 and NFAT2 between Ly-6C$^-$ and Ly-6C$^+$ CD4 T$_N$ cells were diminished but not completely abolished (*Figure 4I* and *Figure 4—figure supplement 2*).

Altogether, our data demonstrate that increasing in vitro intracellular Ca$^{2+}$ levels is sufficient to down-modulate Ly-6C expression at the cell surface of Ly-6C$^+$ CD4 T$_N$ cells without inducing any significant proliferation or activation. Accordingly, the greater nuclear-cytoplasmic ratio of NFAT proteins observed in Ly-6C$^-$ CD4 T$_N$ cells, when compared to their Ly-6C$^+$ CD4 T$_N$-cell counterparts, might reflect differences in the intensity of the Ca$^{2+}$/Calcineurin signaling induced in vivo in these cells. Such differences could result from the differential ability of these cells to regularly interact with self-MHC/self-peptide complexes in the steady-state.

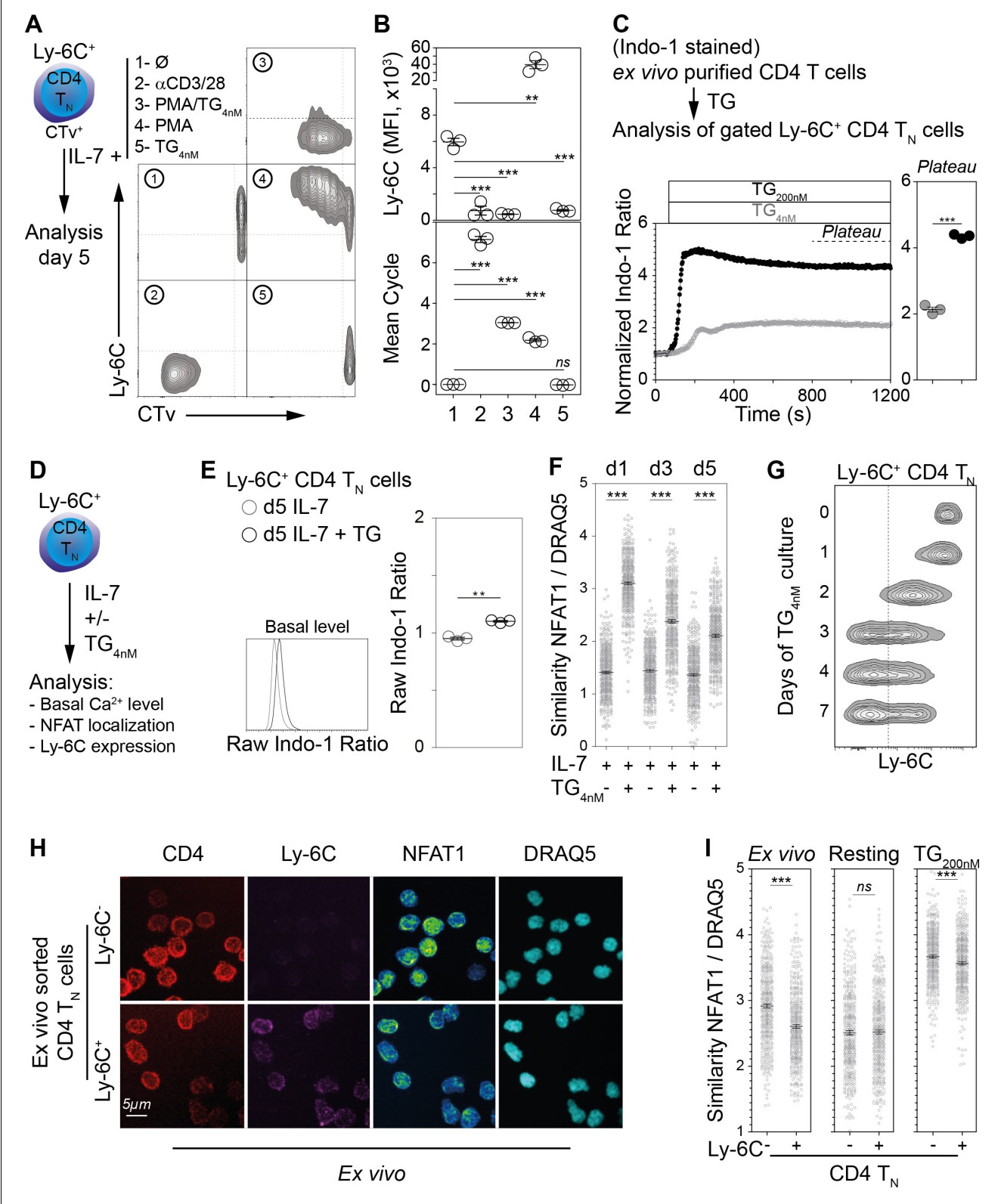

**Figure 4.** Ly-6C⁻ CD4 T_N-cell phenotype relies upon calcium signaling pathway in vitro. (A–B) Flow-cytometry sorted Ly-6C⁺ CD4 T_N cells from C57BL/6 Foxp3-GFP mice were labelled with CTv and cultured in IL-7 (10 ng/ml) in the presence of either coated αCD3/28 (4µg/ml), PMA and TG (1.25 ng/ml and 4 nM, respectively), PMA alone (1.25 ng/ml) or TG (4 nM). Cells were recovered and analyzed after 5 days of culture. (A) Representative Ly-6C/CTv contour-plots are shown. (B) Ly-6C Mean fluorescence intensities (MFIs), for gated CD4⁺ TCR⁺ cells, are shown as means ± s.e.m. for a representative

*Figure 4 continued on next page*

*Figure 4 continued*

experiment with three mice per group (upper panel). Mean cell cycle numbers were calculated (lower panel). (C) Ex-vivo-purified CD4 T cells from C57BL/6 Foxp3-GFP mice were stained with Indo-1 and cell surface molecules CD44, Ly-6C and lineage markers (CD11c, CD11b, CD8β, CD25, TCRγδ and NK1.1). Intracellular calcium concentration was assessed before and after stimulation with 4 or 200 nM TG to the extracellular medium and monitored by flow cytometry for 20 min; results are presented as normalized ratio of Indo-1 emission at 405 nm to that at 510 nm (405/510) for gated Ly-6C$^+$ CD4 T$_N$ cells (Lineage$^-$ Foxp3-GFP$^-$ CD44$^{lo}$ Ly-6C$^+$ cells). Normalized Indo-1 ratio at the *Plateau* are shown as means ± s.e.m. for a representative experiment (out of two independent experiments) with three mice per group (Each dot represents an individual mouse). (D–G) Flow-cytometry sorted Ly-6C$^+$ CD4 T$_N$ cells from C57BL/6 Foxp3-GFP mice were cultured in the presence of IL-7 (10 ng/mL) with TG (4 nM) or not. (D) Diagram illustrating the experimental protocol. (E) Basal intracellular calcium concentration was assessed, as in C, after 5 days of culture. Raw Indo-1 ratio are shown as means ± s.e.m. for a representative experiment (out of two independent experiments) with three mice per group (each dot represents an individual mouse). (F) After 1, 3 and 5 days of culture, cells were analyzed by imaging flow cytometry. NFAT1 nuclear localization was calculated as similarity score between NFAT1 and DRAQ5 intensities. Data are representative of one of two independent experiments. (G) Cells were analyzed after 0, 1, 2, 3, 4 and 7 days of culture. Representative Ly-6C contour plots are shown for gated CD4 T$_N$ cells (CD4$^+$ CD44$^{lo}$ CD25$^{lo}$ Foxp3-GFP$^-$) are shown. (H, I) LN cells were isolated from C57BL/6 mice and fixed in 4% paraformaldehyde immediately (Ex vivo) or after 30 min of culture in the presence of 200 nM of TG (TG) or not (Resting). Ly-6C$^-$ and Ly-6C$^+$ CD4 T$_N$ cells (CD4$^+$ CD44$^{lo}$ CD25$^{lo}$ Foxp3$^-$) were sorted by flow cytometry and stained for NFAT1, and DNA (DRAQ5). (H) Cells were analyzed by confocal microscopy; CD4 (Red), Ly-6C (Magenta), NFAT1 (pseudocolor) and DNA (DRAQ5, cyan) fluorescence are shown for 'ex vivo' purified Ly-6C$^-$ (upper panel) and Ly-6C$^+$ (lower panel) CD4 T$_N$ cells. Original magnification ×63. (I) Cells were analyzed by imaging flow cytometry and NFAT1 nuclear localization assessed as in F. Data are representative of one of three independent experiments. (B, C, E, F, I) Significance of differences were assessed using a two-tailed unpaired Student's t-test. Values of p<0.05 were considered as statistically significant (**p<0.01; ***p<0.001; *ns*, not significant).

DOI: https://doi.org/10.7554/eLife.27215.008

The following figure supplements are available for figure 4:

**Figure supplement 1.** 'Ca$^{2+}$-converted' Ly-6C$^+$CD4 T$_N$ cells keep a naive phenotype.

DOI: https://doi.org/10.7554/eLife.27215.009

**Figure supplement 2.** NFAT2 localization is more nuclear in Ly-6C$^-$ CD4 T$_N$ cells than in their Ly-6C$^+$-cell counterparts.

DOI: https://doi.org/10.7554/eLife.27215.010

## The Ca$^{2+}$-calcineurin signaling pathway shapes the phenotype of the CD4 T$_N$-cell compartment

We have identified several proteins differentially expressed between Ly-6C$^-$ and Ly-6C$^+$ CD4 T$_N$ cells (*Figure 2—figure supplement 1*) and have showed that TG induced Ly-6C downregulation at the cell surface of Ly-6C$^+$ CD4 T$_N$ cells. We next studied whether proteins of the 6CSign other than Ly-6C, were also modulated by an increase in intracellular Ca$^{2+}$. To go further, we examined in parallel the involvement of the calcineurin phosphatase in these processes. To this aim, Ly-6C$^+$ CD4 T$_N$ cells were isolated from peripheral LNs of C57BL/6 Foxp3-GFP mice by flow cytometry sorting and cultured with IL-7 in the presence or absence of TG and calcineurin-inhibitors (Cyclosporin A, CsA and Tacrolimus, FK506, FK). Ly-6C$^-$ CD4 T$_N$ cells cultured in IL-7 were added as control. After 5 days of culture in these conditions, the expression of Ly-6C, CD5, CD73, CD122, CD200 and Izumo1r was analyzed by flow-cytometry (*Figure 5A*). For all these proteins, TG induced changes in their expression at the cell surface of Ly-6C$^+$ CD4 T$_N$ cells. More precisely, their level of expression reached those observed in Ly-6C$^-$ CD4 T$_N$ cells. Blocking calcineurin activation with either CsA or FK506 led to the complete inhibition of this phenotypic conversion of Ly-6C$^+$ (CD5$^{lo}$, CD73$^{int}$, CD122$^{lo}$, CD200$^{lo}$, Izumo1r$^{lo}$) CD4 T$_N$ cells into Ly-6C$^-$ (CD5$^{hi}$, CD73$^{hi}$, CD122$^{int}$, CD200$^{int}$, Izumo1r$^{hi}$) CD4 T$_N$ cells. This Ca$^{2+}$-induced phenotypic conversion thus depends on the activity of the canonical Ca$^{2+}$-calcineurin signaling pathway.

We then investigated whether this in vitro observation could be mimicked in vivo. We first confirmed that the Ca$^{2+}$-calcineurin signaling cascade was active in vivo in Ly-6C$^-$ CD4 T$_N$ cells by showing that blocking calcineurin activation for 18 hr with FK506 was sufficient to abrogate the nuclear localization of NFAT in these cells (*Figure 5—figure supplement 1*). We then wondered whether a longer treatment with this calcineurin inhibitor would affect the phenotype of CD4 T$_N$ cells in vivo. To this aim, C57BL/6 Foxp3-GFP mice were injected daily with FK506 or PBS for 2 weeks (*Figure 5B*). After 14 days, CD4 T$_N$ cells from peripheral LNs and the spleen were analyzed for their expression of Ly-6C, Izumo1r and CD200. In line with our in vitro experiments, both the percentage of Ly-6C$^+$ cells among CD4 T$_N$ cells and the MFI of Ly-6C at the cell surface of Ly-6C$^+$ CD4 T$_N$ cells increased in treated mice when compared to control mice (*Figure 5C–E*). Moreover, FK506 induced significant decreases of Izumo1r and CD200 surface levels in CD4 T$_N$ cells (*Figure 5C,F*). Such changes in the

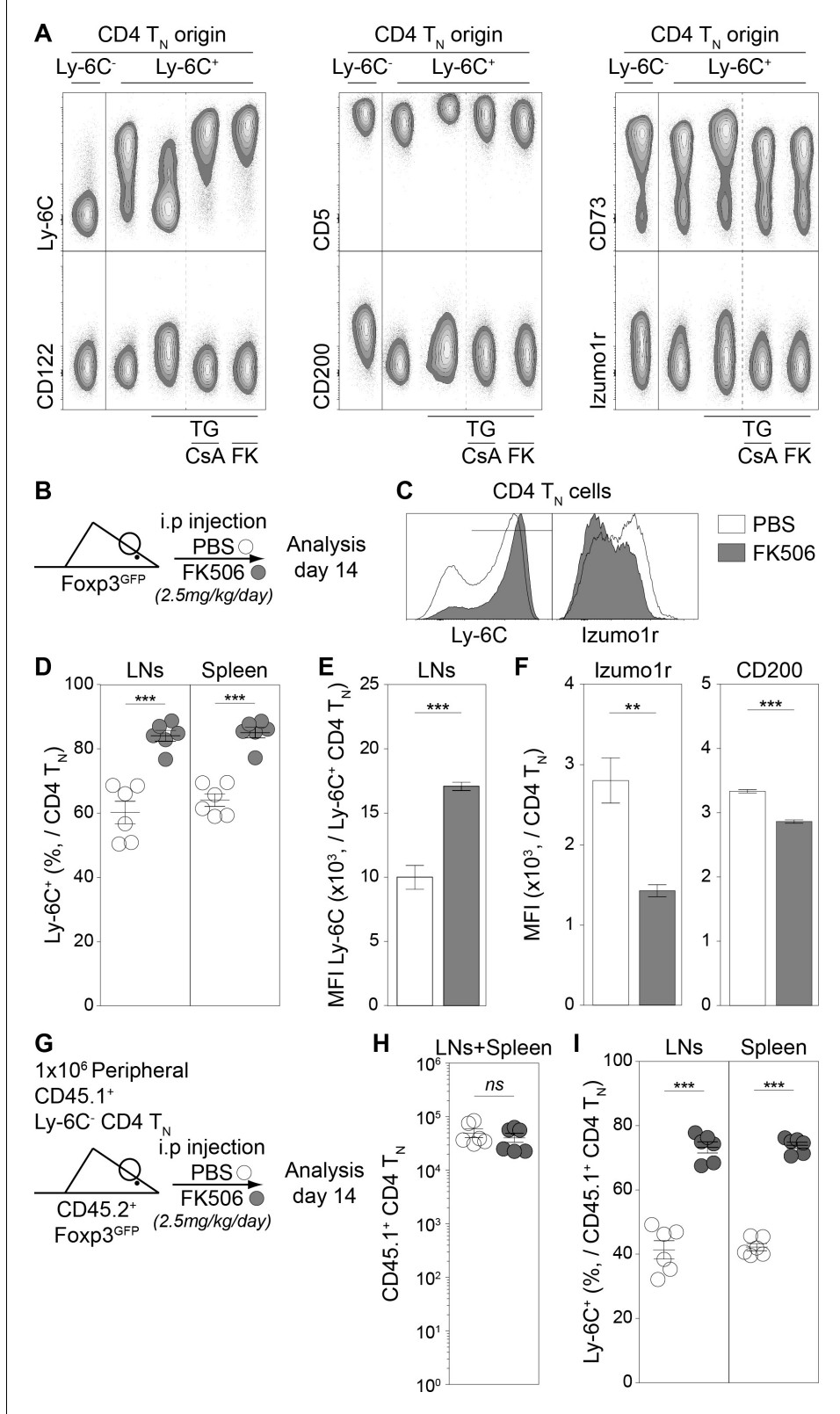

**Figure 5.** The calcium-calcineurin pathway shapes the phenotype of the CD4 $T_N$-cell compartment in vivo. (**A**) Flow-cytometry sorted Ly-6C$^+$ CD4 $T_N$ cells from C57BL/6 Foxp3-GFP mice were cultured in IL-7 (10 ng/mL) alone or in the presence of either TG (4 nM), TG and Cyclosporin A (CsA; 50 nM) or TG and FK506 (FK; 200 nM). Flow-cytometry sorted Ly-6C$^-$ CD4 $T_N$ cells rested in IL-7 were used as control. After 5 days, cells were analyzed for their expression of Ly-6C, CD5, CD73, CD122, CD200 and Izumo1r. Representative contour-plots of cell surface markers are shown for gated CD4 $T_N$ cells

*Figure 5 continued on next page*

*Figure 5 continued*

(CD4$^+$ TCRβ$^+$ CD44$^{lo}$ CD25$^{lo}$ Foxp3-GFP$^-$) as a function of culture condition. (**B–F**) C57BL/6 Foxp3-GFP mice were daily injected intraperitoneally with Prograf (FK506; 2.5 mg/kg) or diluent (PBS). Two weeks after treatment LNs (pooled pLNs and mLNs) and spleen were recovered and CD4 T cells were analyzed. (**B**) Diagram illustrating the experimental procedure. (**C**) Ly-6C and Izumo1r fluorescence histograms for gated CD4 T$_N$ cells (CD4$^+$ TCRβ$^+$ CD44$^{lo}$ CD25$^{lo}$ Foxp3-GFP$^-$) recovered from LNs of PBS (white) and FK506 (grey) treated mice. (**D**) Percentage of Ly-6C$^+$ cells among CD4 T$_N$ (CD4$^+$ TCRβ$^+$ CD44$^{lo}$ CD25$^{lo}$ Foxp3-GFP$^-$) cells are shown for LNs and spleens of PBS (white) and FK506 (grey) treated mice. (**E**) Ly-6C Mean fluorescence intensities (MFIs), for gated Ly-6C$^+$ CD4 T$_N$ (Ly-6C$^+$ CD4$^+$ TCRβ$^+$ CD44$^{lo}$ CD25$^{lo}$ Foxp3-GFP$^-$) cells recovered from LNs of PBS (white) and FK506 (grey) treated mice, are shown as means ± s.e.m. for two independent experiments with three mice per group. (**F**) Izumo1r and CD200 mean fluorescence intensities (MFIs), for gated Ly-6C$^+$ CD4 T$_N$ (Ly-6C$^+$ CD4$^+$ TCRβ$^+$ CD44$^{lo}$ CD25$^{lo}$ Foxp3-GFP$^-$) cells recovered from LNs of PBS (white) and FK506 (grey) treated mice, are shown as means ± s.e.m. for a representative experiment with three mice per group. (**G–I**) $1 \times 10^6$ flow-cytometry sorted Ly-6C$^-$ CD4 T$_N$ cells from CD45.1$^+$ C57BL/6 Foxp3-GFP mice were adoptively transferred into sex-matched CD45.2$^+$ C57BL/6 Foxp3-GFP recipient mice daily injected intraperitoneally with Prograf (FK506; 2.5 mg/kg) or diluent (PBS). Two weeks after transfer and treatment, LNs (pooled pLNs and mLNs) and spleen were recovered and donor-derived CD45.1$^+$ CD4 T cells were analyzed. (**G**) Diagram illustrating the experimental model. (**H**) Absolute numbers of donor-derived CD4 T$_N$ (CD45.1$^+$ CD45.2$^-$ CD4$^+$ TCRβ$^+$ CD44$^{lo}$ CD25$^{lo}$ Foxp3-GFP$^-$) cells recovered from LNs and spleen of recipient mice are shown as means ± s.e.m. for two independent experiments with three mice per group. (**I**) Percentage of Ly-6C$^+$ among donor-derived CD4 T$_N$ (CD45.1$^+$ CD45.2$^-$ CD4$^+$ TCRβ$^+$ CD44$^{lo}$ CD25$^{lo}$ Foxp3-GFP$^-$) cells recovered from LNs and spleen of recipient mice are shown as means ± s.e.m. for two independent experiments with three mice per group. (**D, H, I**) Each dot represents an individual mouse. (**D-F; H, I**) Significance of differences were assessed using a two-tailed unpaired Student's t-test. Values of p<0.05 were considered as statistically significant (**p<0.01; ***p<0.001; *ns*, not significant).

DOI: https://doi.org/10.7554/eLife.27215.011

The following figure supplement is available for figure 5:

**Figure supplement 1.** The calcium-calcineurin cascade drives NFAT nuclear translocation in CD4 T$_N$ cells in vivo.

DOI: https://doi.org/10.7554/eLife.27215.012

phenotype of the bulk of CD4 T$_N$ cells could result from either the conversion of Ly-6C$^-$ CD4 T$_N$ cells into Ly-6C$^+$ CD4 T$_N$ cells or the disappearance of the Ly-6C$^-$-cell subset. We therefore decided to compare the behavior of adoptively transferred Ly-6C$^-$ CD4 T$_N$ cells in FK506 or PBS-treated mice (*Figure 5G*). $10^6$ Ly-6C$^-$ CD4 T$_N$ cells purified from LNs of CD45.1$^+$ Foxp3-GFP donor mice were adoptively transferred into CD45.2$^+$ Foxp3-GFP-recipient mice. Host mice were then daily injected with FK506 or PBS for 2 weeks (*Figure 5G*). After 14 days, donor-derived CD4 T$_N$ cells from peripheral LNs and the spleen were analyzed. Although similar numbers of donor-derived CD4 T$_N$ cells were recovered from both FK506 and PBS -treated mice (*Figure 5H*), these cells were still greatly enriched in Ly-6C-expressing cells in FK506-treated recipients (*Figure 5I*).

Altogether, our data demonstrate that the activation of the Ca$^{2+}$-calcineurin signaling pathway drives the phenotypic conversion of Ly-6C$^+$ CD4 T$_N$ cells into Ly-6C$^-$ CD4 T$_N$ cells both in vitro and in vivo.

## Ca$^{2+}$-mediated shaping of the CD4 T$_N$-cell iTreg-cell differentiation potential

As a rise in intracellular Ca$^{2+}$ level converts phenotypically Ly-6C$^+$ CD4 T$_N$ cells into Ly-6C$^-$ CD4 T$_N$ cells, we then tested whether the in vitro iTreg-cell polarization potential of these ex-Ly-6C$^+$ CD4 T$_N$ cells (referred thereafter as 'Ca$^{2+}$-converted' Ly-6C$^+$ CD4 T$_N$ cells) was also modified. Ly-6C$^-$ and Ly-6C$^+$ CD4 T$_N$ cells were recovered from C57BL/6 Foxp3-GFP mice and cultured in vitro with or without TG. After 5 days of culture, viable cells were FACS-sorted and stimulated with αCD3- and αCD28-coated antibodies in the presence of graded doses of TGFβ for 4 days (*Figure 6A*). Of note, even after 5 days of resting in the presence of IL-7, Ly-6C$^-$ CD4 T$_N$ cells were keeping a greater sensitivity to iTreg-cell polarization signals, than Ly-6C$^+$ CD4 T$_N$ cells cultured in the same conditions. Importantly, the iTreg-cell polarization potential of Ly-6C$^+$ CD4 T$_N$ cells rose up when these cells were pre-incubated in the presence of TG and became similar to the one observed for Ly-6C$^-$ CD4 T$_N$ cells (*Figure 6B,C*). In agreement with the fact that calcineurin inhibitors blocked the TG-mediated phenotypic conversion of Ly-6C$^+$ CD4 T$_N$ cells into Ly-6C$^-$ CD4 T$_N$ cells (*Figure 5A*), adding CsA at the time of TG pre-incubation also abrogated the sensitization of Ly-6C$^+$ CD4 T$_N$ cells to iTreg-cell polarization signals (*Figure 6B*). EC50 values for TGFβ were calculated in these conditions and were statistically different between the 2 CD4 T$_N$-cell subsets when cells were pre-incubated in IL-7 alone but dropped to similar levels when TG was added in the pre-culture medium (*Figure 6C*). Of note, pre-incubating Ly-6C$^+$ CD4 T$_N$ cells with TG and CsA further limit their ability to commit

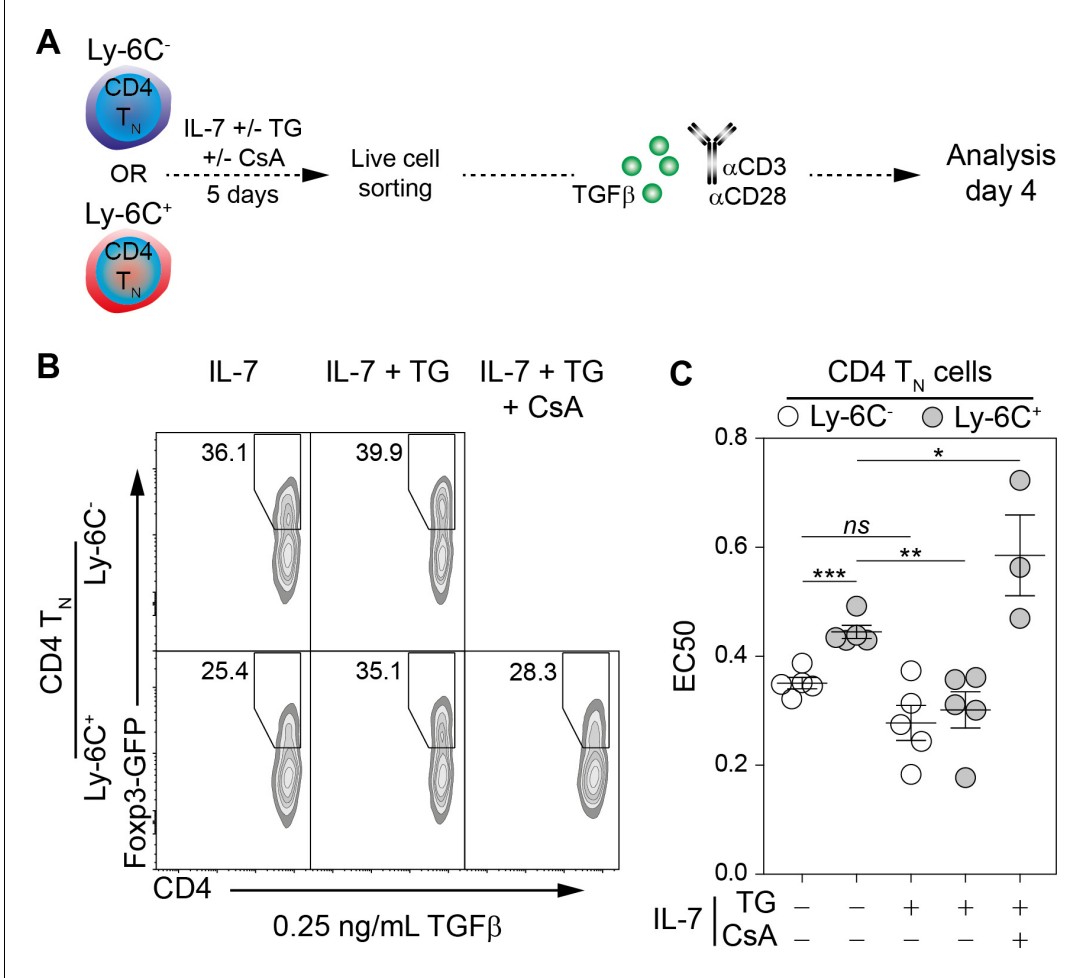

**Figure 6.** Ly-6C⁻ CD4 $T_N$-cell sensitization to iTreg-cell polarization signals relies upon calcium signaling pathway in vitro. Flow-cytometry sorted Ly-6C⁻ and Ly-6C⁺ CD4 $T_N$ cells from C57BL/6 Foxp3-GFP mice were cultured in IL-7 (10 ng/mL) with or without TG (4 nM) and CsA (50 nM). After 5 days, live cells were flow-cytometry sorted and stimulated with coated αCD3 and αCD28 Abs (4 µg/ml) in the presence of graded doses of TGFβ1. Cells were analyzed after 4 days of stimulation. (**A**) Diagram illustrating the experimental procedure. (**B**) Representative Foxp3/CD4 contour-plots and proportions of Foxp3⁺ cells for gated CD4⁺ cells are shown at a dose of 0.25 ng/mL TGFβ1, as a function of pre-culture condition. (**C**) Concentrations of TGFβ1 needed to obtain 50% of the maximal percentage of iTreg-cell polarization (EC50) were calculated for each CD4 $T_N$-cell subset and each pre-culture condition. Each dot represents an independent experiment. Significance of differences were assessed using a two-tailed paired Student's t-test. Values of $p < 0.05$ were considered as statistically significant (*$p < 0.05$; **$p < 0.01$; ***$p < 0.001$; *ns*, not significant).
DOI: https://doi.org/10.7554/eLife.27215.013

into iTreg cells as reflected by a significant increase in EC50 (*Figure 6C*). Altogether, our data demonstrate that an increase in intracellular $Ca^{2+}$ levels not only shapes the phenotype of the CD4 $T_N$-cell compartment but also sensitizes in vitro these cells to iTreg-cell polarization signals, both processes occurring through a calcineurin-dependent pathway.

To confirm these data in vivo, we used the well-known model of antigen-specific pTreg-cell development induced by oral tolerance (*Coombes et al., 2007*; *Sun et al., 2007*). This protocol studies the behavior of CD4 $T_N$ cells from ovalbumin-specific TCR transgenic OT-II mice adoptively transferred into wild-type mice fed with ovalbumin (OVA). Indeed, in these conditions, a significant proportion of OT-II cells rapidly differentiate into pTreg cells in the mesenteric lymph nodes and Peyer Patches of recipient mice. Most OT-II CD4 $T_N$ cells expressed Ly-6C ex vivo (*Figure 7—figure supplement 1A*). FACS-sorted CD45.1/2⁺ OT-II CD4 $T_N$ cells were first cultured in IL-7 in the presence or absence of TG (*Figure 7A*). After 5 days of culture, TG led to a marked downregulation of Ly-6C (*Figure 7—figure supplement 1B*). Living cells were then FACS-sorted and $0.5–1.10^6$ cells were adoptively transferred into CD45.1 Foxp3-GFP mice. Finally, recipient mice were fed or not for 7

days with OVA in their drinking water. As expected, OVA administration led to the activation of OT-II cells, as reflected by a significant CD44 upregulation at their cell surface (*Figure 7—figure supplement 1C*). Similar numbers of OT-II CD4 T cells were recovered from the secondary lymphoid organs of OVA-fed mice whether they were initially injected with 'Ca$^{2+}$-converted' or not OT-II CD4 $T_N$ cells (*Figure 7B*). In all secondary lymphoid organs, Ca$^{2+}$-converted OT-II CD4 $T_N$ cells gave rise to greater proportions and absolute numbers of Foxp3-expressing cells than OT-II CD4 $T_N$ cells cultured with IL-7 alone prior to injection (*Figure 7C–E*). Specifically, a total of $1.16 \pm 0.22 \times 10^4$ pTreg (Foxp3-expressing) cells were recovered from the whole periphery of recipient mice injected with Ly-6C$^+$ OT-II CD4 $T_N$ cells compared to $2.26 \pm 0.32 \times 10^4$ pTreg cells when mice were injected with Ca$^{2+}$-converted OT-II CD4 $T_N$ cells ($p<0.05$).

These latter results were confirmed by using a second protocol of oral administration of OVA. In this setting, 'Ca$^{2+}$-converted' OT-II CD4 $T_N$ cells were co-transferred with OT-II CD4 $T_N$ cells cultured in IL-7 alone in order to compare their ability to convert into pTreg in the same recipient mice. FACS-sorted CD45. 2$^+$ and CD45.1/2$^+$ OT-II CD4 $T_N$ cells were first cultured in IL-7 in the absence or presence of TG, respectively (*Figure 7—figure supplement 1A*). After 5 days of culture, living cells were FACS-sorted, mixed at a 1:1 ratio and $1.10^6$ cells were adoptively transferred into CD45.1 Foxp3-GFP mice. Finally, recipient mice were fed with OVA by gavage (4 and 24 hr after the transfer of OT-II cells). Nine days later, secondary lymphoid organs were recovered and the phenotype of donor-derived OT-II T cells was analyzed. In this setting, 'Ca$^{2+}$-converted' OT-II CD4 $T_N$ cells were also giving rise to greater absolute numbers of Foxp3-expressing cells than OT-II CD4 $T_N$ cells cultured with IL-7 alone prior to injection (*Figure 7—figure supplement 1B,C*). More precisely, more than three quarters of the Foxp3$^+$ OT-II cells arising in these conditions derived from 'Ca$^{2+}$-converted' cells (*Figure 7—figure supplement 1D*).

These latter results validate our in vitro data showing that a rise in intracellular Ca$^{2+}$ leads to an enhanced sensitivity of CD4 $T_N$ cells to iTreg-cell polarization signals.

## Discussion

In the steady-state, naive T cells continually recirculate between the blood, lymph and secondary lymphoid organs, scanning dendritic cells (DCs) for the presence of foreign antigens. In the course of their journey, naive T cells also make weak, but functional, interactions with self-peptides presented by self-MHC molecules (self-MHC). Such contacts with self-MHC are required for the long-term survival of peripheral naive T cells (*Martin et al., 2006*, *2003*; *Stritesky et al., 2012*; *Tanchot et al., 1997*). The signals derived from the recognition of self-MHC by TCRs also allow maintaining naive T cells in a state of greater sensitivity for responses to foreign antigens (*Dorfman et al., 2000*; *Stefanová et al., 2002*). The seminal work of Štefanová et al. showed a rapid decline in the ability of CD4 $T_N$ cells to respond to their cognate antigen once contacts with self-MHC were disrupted (*Stefanová et al., 2002*). These findings were confirmed by several groups using various elegant experimental models (*Hochweller et al., 2010*; *Lo et al., 2009*; *Mandl et al., 2013*; *Persaud et al., 2014*). Beside these works, we have recently demonstrated that CD4 $T_N$-cell self-reactivity not only increases quantitatively but also shapes qualitatively their response toward their cognate antigens in the effector phase by increasing their ability to commit toward the iTreg/pTreg-cell lineage (*Martin et al., 2013*). In the present paper, we first wondered whether the enhanced ability of the most self-reactive CD4 $T_N$ cells to convert into iTreg/pTreg cells upon appropriate stimulation was a cell-intrinsic property. The unchanged ability of both Ly-6C$^-$ and Ly-6C$^+$ CD4 $T_N$ cells to polarize into Foxp3-expressing iTreg cells in vitro whether they were cultured together or separately demonstrate that the biased commitment of the most self-reactive CD4 $T_N$ cells toward the iTreg-cell lineage is cell-intrinsic.

We have recently described the cell surface GPI-anchored protein, Ly-6C, as an additional and complementary sensor of T-cell self-reactivity (*Martin et al., 2013*). However, significant differences may be noticed between CD5 and Ly-6C. First, whereas CD5 surface levels directly correlate with self-reactivity, Ly-6C expression by peripheral CD4 $T_N$ cells inversely correlates with their ability to interact with self-MHC. Second, in contrast to CD5, Ly-6C expression at the cell surface of CD4 $T_N$ cells is stable over time in homeostatic conditions as its up-regulation after self-MHC deprivation takes several days (*Martin et al., 2013*). Notwithstanding these differences, Ly-6C$^-$ CD4 $T_N$ cells express higher protein and mRNA levels of CD5 than their Ly-6C$^+$-cell counterparts. Two recent

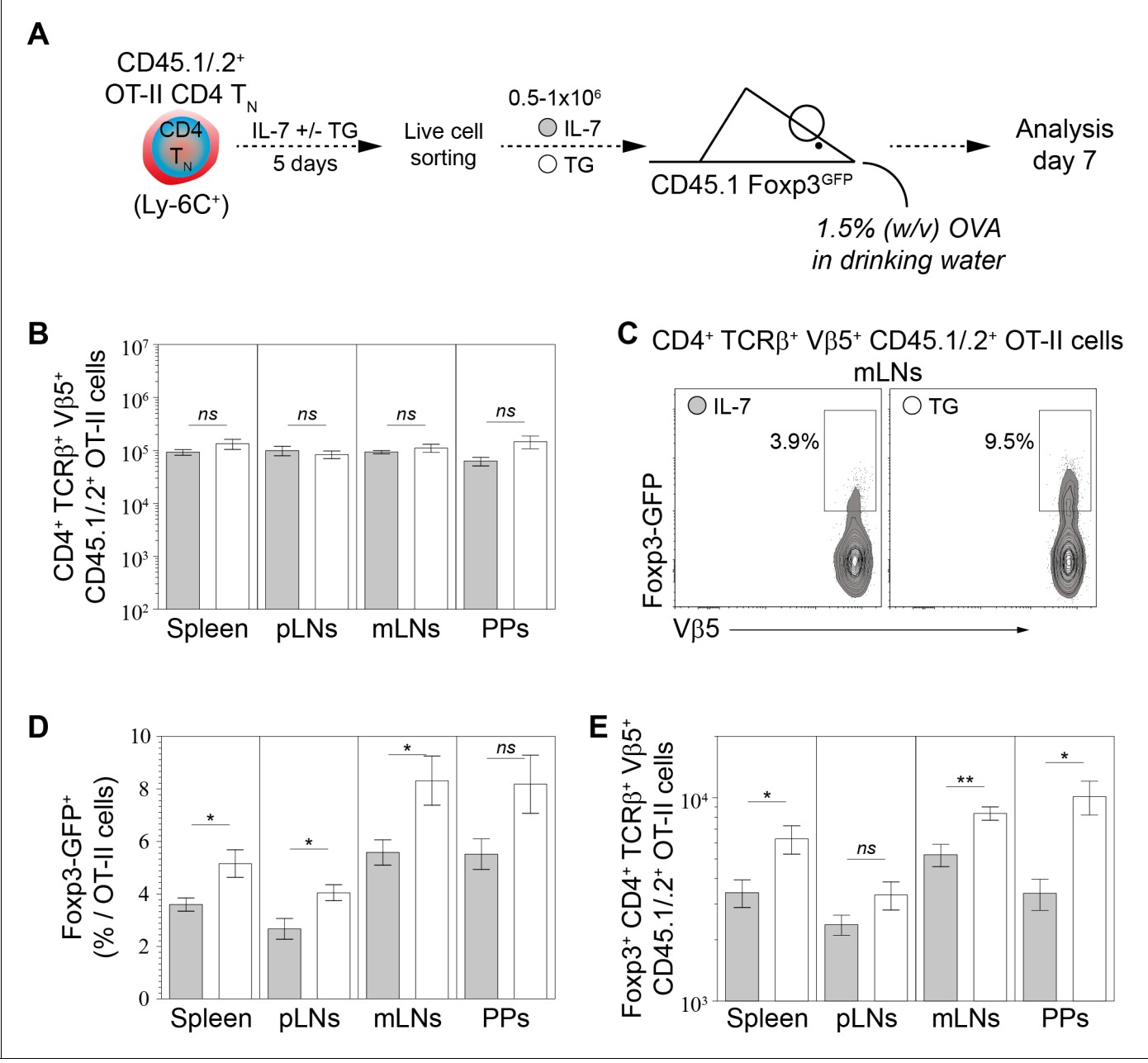

**Figure 7.** Calcium-mediated shaping of the CD4 $T_N$-cell pTreg-cell differentiation potential in vivo. Purified CD4 T cells from CD45.1/.2$^+$ C57BL/6 Foxp3-GFP OT-II mice were cultured in IL-7 (10 ng/ml) with or without TG (4 nM). After 5 days live CD4 $T_N$ (CD44$^{lo}$ CD25$^{lo}$ CD8β$^-$ CD11b$^-$ CD11c$^-$ NK1.1$^-$ TCRγδ$^-$ Foxp3-GFP$^-$) cells were flow-cytometry sorted and injected intravenously (0.5−1 × 10$^6$ cells) into sex-matched CD45.1$^+$ C57BL/6 Foxp3-GFP recipient mice fed with Ovalbumin (OVA; 1.5% w/v) in the drinking water. One week after transfer, peripheral and mesenteric LNs (pLNs and mLNs, respectively), Peyer's Patches (PPs) and spleen were recovered separately and donor-derived CD4 T cells were analyzed. (A) Diagram illustrating the experimental model. (B) Absolute numbers of donor-derived OT-II CD4 T (CD45.1$^+$ CD45.2$^+$ CD4$^+$ TCRβ$^+$ Vβ5$^+$) cells recovered from pLNs, mLNs, PPs and spleen of recipient mice are shown as means ± s.e.m. for three independent experiments with two or three mice per group. (C) Representative Foxp3/Vβ5 contour-plots and proportions of Foxp3-GFP$^+$ cells for gated donor-derived OT-II CD4 T (CD45.1$^+$ CD45.2$^+$ CD4$^+$ TCRβ$^+$ Vβ5$^+$) cells recovered from mLNs are shown. (D–E) Percentages (D) and absolute numbers (E) of Foxp3-GFP$^+$ among donor-derived OT-II CD4 T (CD45.1$^+$ CD45.2$^+$ CD4$^+$ TCRβ$^+$ Vβ5$^+$) cells recovered from pLNs, mLNs, PPs and spleen of recipient mice are shown as means ± s.e.m. for three independent experiments with two or three mice per group. (B, D, E) Significance of differences were assessed using a two-tailed unpaired Student's t-test. Values of p<0.05 were considered as statistically significant (*p<0.05; *ns*, not significant). FIGURE SUPPLEMENT LEGENDS.

DOI: https://doi.org/10.7554/eLife.27215.014

The following figure supplements are available for figure 7:

*Figure 7 continued on next page*

*Figure 7 continued*

**Figure supplement 1.** OT-II CD4 $T_N$ cells are Ly-6C$^+$ and can be Ca$^{2+}$-converted and tested for their ability to convert into pTreg cells in vivo.
DOI: https://doi.org/10.7554/eLife.27215.015

**Figure supplement 2.** Calcium-mediated shaping of the CD4 $T_N$-cell phenotype increases the pTreg-cell differentiation potential of OT-II CD4 $T_N$ cells in vivo.
DOI: https://doi.org/10.7554/eLife.27215.016

papers by the group of Daniel Hawiger have highlighted a crucial role of CD5 in promoting the conversion of CD4 $T_N$ cells into iTreg/pTreg cells (*Henderson et al., 2015*; *Jones et al., 2016*). CD5 would block the activation of the mammalian target of rapamycin (mTOR) and would allow activated CD4 $T_N$ cells to resist to the inhibition of iTreg-cell induction induced by $T_H$1- and $T_H$2-cell-derived cytokines. Accordingly, in the absence of effector-differentiating cytokines, CD5$^{hi}$ and CD5$^{lo}$ CD4 $T_N$ cells were shown to differentiate with a similar efficiency into iTreg/pTreg cells (*Henderson et al., 2015*). Such a phenomenon is unlikely to account for the greater ability of Ly-6C$^-$ CD4 $T_N$ cells to commit to the iTreg/pTreg-cell lineage. Indeed, Ly-6C$^-$ and Ly-6C$^+$ CD4 $T_N$ cells produced similar amounts of these cytokines after stimulation and Ly-6C$^-$ CD4 $T_N$ cells still differentiated more efficiently than their Ly-6C$^+$-cell counterparts into iTreg cells in vitro in the presence of anti-cytokine (IL-4 and IFN-γ) blocking antibodies (*Figure 1—figure supplement 1C–E*). Moreover, whereas rapamycin drastically diminished the difference in the ability of CD5$^{hi}$ and CD5$^{lo}$ CD4 $T_N$ cells to convert to iTreg/pTreg cells in the presence of cytokines known as restraining this effector fate (*Henderson et al., 2015*), this mTOR inhibitor similarly enhanced the generation of iTreg cells from both Ly-6C$^-$ and Ly-6C$^+$ CD4 $T_N$ cells and thus preserved the difference between these two cell subsets (data not shown).

In the present study, we have identified the Ca$^{2+}$ signaling pathway as sufficient to induce Ly-6C down-regulation at the cell surface of CD4 $T_N$ cells in vitro. Indeed, incubation of Ly-6C$^+$ CD4 $T_N$ cells with the sarco/endoplasmic reticulum calcium ATPase inhibitor, thapsigargin, led to multiple phenotypic changes including not only Ly-6C down-regulation but also variations in the expression of many other genes of the 6CSign (such as CD5, CD73, CD122, CD200 and Izumo1r). This phenotypic conversion of Ly-6C$^+$ CD4 $T_N$ cells into Ly-6C$^-$ CD4 $T_N$ cells takes four days to occur in vitro and relies on the activity of Calcineurin, as shown by its complete blocking in the presence of Cyclosporin A or FK506. Interestingly, calcium- and PKC/Ras-dependent signaling pathways had divergent effects on the expression of Ly-6C. Indeed, whereas TG induced Ly-6C down-regulation, PMA led to its upregulation (*Figure 4A*). In line with this observation, the 6CSign does not correlate with the changes in gene expression induced by PMA (*Figure 3B*). These opposite effects of TG and PMA may reflect the well-documented and complex interplay between the PKC and Ca$^{2+}$ signaling pathways. For example, PKC translocation to the plasma membrane is strictly Ca$^{2+}$ dependent (*Reither et al., 2006*) and calcineurin is phosphorylated and inhibited by PKC (*Hashimoto and Soderling, 1989*). Altogether, our results suggest that interactions with self-MHC in the steady-state result in a dominant Ca$^{2+}$ signaling (when compared to PKC and Ras-dependent pathways) leading to down-regulation of Ly-6C expression. This hypothesis is consistent with our results showing that in vivo Calcineurin inhibition leads to an increase in Ly-6C expression at even higher levels than those observed at the cell surface of Ly-6C$^+$ CD4 $T_N$ cells from untreated mice. This hypothesis is reinforced by the work of Dong et al. (Dong TX et al., co-published with the present article) showing that Ca$^{2+}$ fluxes can be measured in mouse total lymph node T cells in the steady-state and that anti-MHC blocking antibodies significantly reduced their frequency. Continuous interactions with self-MHC in the steady-state may thus induce calcium waves that shape both the phenotype of CD4 $T_N$ cells and their behavior in the effector phase by favoring their differentiation into pTreg cells.

tTreg and pTreg cells have complementary roles in immune-mediated tolerance (*Haribhai et al., 2011*). An attractive hypothesis would be that tTreg cells would be responsible for tolerance to self-antigens, whereas pTreg cells would be in charge of restraining deleterious immune responses to non-self-antigens. In particular, pTreg cells are involved in the control of the responses to non-self-antigens leading to allergy and asthma (*Josefowicz et al., 2012*) as well as to commensal organism- (*Lathrop et al., 2011*) and food-derived antigens (*Mucida et al., 2005*) in the gut. Foetus-derived and allograft-derived antigens represent other obvious examples of acute exposure to non-self-antigens arising in the adults and requiring the establishment of a tolerance. In both cases, pTreg cells

are generated against non-self antigens (either conceptus-male-derived [*Samstein et al., 2012*] or allograft-derived [*Francis et al., 2011*; *Wood et al., 2012*]). These cells are needed to establish an efficient tolerance toward the foetus (*Samstein et al., 2012*). However, there is still a lack of evidence to definitely implicate pTreg cells in the induction of an efficient tolerance toward allograft, in part because of the difficulties to achieve such a state. We have previously demonstrated that self-reactivity in the steady-state increases the ability of CD4 $T_N$ cells to differentiate into iTreg/pTreg cells (*Martin et al., 2013*). Accordingly, the most self-reactive CD4 $T_N$ cells (i.e. Ly-6C$^-$ CD4 $T_N$ cells) should contribute predominantly to the pTreg-cell pool generated under physiologic and pathologic conditions. In the present study, our data suggest strongly that this tonic TCR-signaling-mediated shaping of the CD4 $T_N$-cell compartment is calcineurin-dependent. In particular, chronic treatment with a calcineurin inhibitor leads to the disappearance of Ly-6C$^-$ CD4 $T_N$ cells. Cyclosporin A and Tacrolimus treatments could thus interfere with the neoconversion of CD4 $T_N$ cells into pTreg cells and limit the development of tolerance in transplant patients. This may explain the difficulty to safely interrupt these immunosuppressive therapies even after years. Thus, besides their obvious clinical utility, calcineurin inhibitors may have potentially harmful side effects that should be studied to better assess and adapt their use.

# Materials and methods

**Key resources table**

| Reagent type (species) or resource | Designation | Source or reference | Identifiers |
|---|---|---|---|
| Strain, strain background (*Mus musculus*) | B6.Cg-Foxp3tm1Mal/J | *Wang et al. (2008)* (PMID: 18209052) | IMSR Cat# JAX:018628, RRID:IMSR_JAX:018628 |
| Strain, strain background (*Mus musculus*) | B6.Cg-Tg(TcraTcrb)425Cbn/J | *Barnden et al., 1998* (PMID: 9553774) | IMSR Cat# JAX:004194, RRID:IMSR_JAX:004194 |
| Antibody | Alexa Fluor 700-conjugated anti CD45.2 (104) | BD Biosciences | Cat# 560693 |
| Antibody | Allophycocyanin (APC)-conjugated anti-CD25 (PC61) | BD Biosciences | Cat# 561048 |
| Antibody | Allophycocyanin (APC)-conjugated anti-CD44 (IM7) | BD Biosciences | Cat# 561862 |
| Antibody | Brilliant Violet (BV) 421-conjugated anti Ly-6C (AL-21) | BD Biosciences | Cat# 562727 |
| Antibody | BV 510-conjugated anti-CD4 (RM4-5) | BD Biosciences | Cat# 563106 |
| Antibody | BV 786-conjugated anti-CD25 (PC61) | BD Biosciences | Cat# 564023 |
| Antibody | Phycoerythrin (PE)-conjugated anti-CD25 (PC61) | BD Biosciences | Cat# 561065 |
| Antibody | Phycoerythrin (PE)-conjugated anti-CD69 (H1.2F3) | BD Biosciences | Cat# 553237 |
| Antibody | Phycoerythrin (PE)-conjugated anti-Izumo1r (TH6) | BD Biosciences | Cat# 560320 |
| Antibody | Phycoerythrin (PE)-conjugated anti-TCRgd (GL3) | BD Biosciences | Cat# 553178 |
| Antibody | Phycoerythrin (PE)-conjugated anti-Vb5.1/5.2 (MR9-4) | BD Biosciences | Cat# 553190 |
| Antibody | PE-Cy7-conjugated anti-CD44 (IM7) | BD Biosciences | Cat# 560569 |
| Antibody | PE-Cy7-conjugated anti-CD45.1 (A20) | BD Biosciences | Cat# 560578 |
| Antibody | Biotinylated anti-CD5 (53–7,3) | BD Biosciences | Cat# 553019 |
| Antibody | Biotinylated anti-CD62L (MEL14) | BD Biosciences | Cat# 553149 |
| Antibody | Biotinylated anti-Ly-6C (AL-21) | BD Biosciences | Cat# 557359 |

*Continued on next page*

*Continued*

| Reagent type (species) or resource | Designation | Source or reference | Identifiers |
|---|---|---|---|
| Antibody | Biotinylated anti-Sca1 (E13-161.7) | BD Biosciences | Cat# 553334 |
| Antibody | Alexa Fluor 647-conjugated anti-IL18ra (BG/IL18ra) | BioLegend | Cat# 132903 |
| Antibody | APC-conjugated streptavidin | BioLegend | Cat# 405207 |
| Antibody | BV 421-conjugated anti-Ly-6C (HK1.4) | BioLegend | Cat# 128032 |
| Antibody | PE-conjugated anti-Ly-6C (HK1.4) | BioLegend | Cat# 128008 |
| Antibody | Alexa 448-conjugated anti-NFAT2 (7A6) | BioLegend | Cat# 649603 |
| Antibody | Alexa 448-conjugated anti-NFAT1 (D43B1) | Cell Signaling | Cat# 14324 |
| Antibody | PE-conjugated anti-CD200 (OX-90) | eBioscience | Cat# 12-5200-82 |
| Antibody | PE-conjugated anti-Ikzf3 (8B2) | eBioscience | Cat# 12-5789-80 |
| Antibody | PE-conjugated anti-Nur77 (12.14) | eBioscience | Cat# 12-5965-82 |
| Antibody | PerCP-Cy5.5-conjugated anti-TCRb (H57-597) | eBioscience | Cat# 45-5961-82 |
| Antibody | Biotinylated anti-CD73 (eBioTY/11.8) | eBioscience | Cat# 14-0731-82 |
| Antibody | APC-conjugated anti-Foxp3 (FJK-165) | eBioscience | Cat# 12-5773-82 |
| Antibody | PE-conjugated anti-Foxp3 (FJK-165) | eBioscience | Cat# 17-5773-82 |
| Antibody | Pacific Blue-conjugated streptavidin | Invitrogen | Cat# S11222 |
| Antibody | APC-Vio770-conjugated anti-CD8a (53–6.7) | Miltenyi Biotec | Cat# 130-102-305 |
| Antibody | PE-conjugated anti-CD122 (TM-b1) | Miltenyi Biotec | Cat# 130-102-569 |
| Chemical compound, drug | Phorbol 12-myristate 13-acetate (PMA) | Calbiochem | CAS 16561-29-8 |
| Chemical compound, drug | FK506 (tacrolimus) | Sigma Aldrich | CAS 109581-93-3 |
| Chemical compound, drug | CellTrace Violet | Invitrogen | Cat# C34557 |
| Chemical compound, drug | CellTrace Far Red | Invitrogen | Cat# C34564 |
| Chemical compound, drug | Recombinant Mouse IL-7 | R and D Systems | Cat# 407 ML-025 |
| Chemical compound, drug | Thapsigargin | Calbiochem | CAS 67526-95-8 |
| Chemical compound, drug | TGFβ1 | Invitrogen | Cat# PHG9204 |
| Chemical compound, drug | DRAQ5 | Cell Signaling | Cat# 4084 |
| Chemical compound, drug | Indo-1, AM | Invitrogen | Cat# I1223 |
| Software, algorithm | Illustrator CS5 | Adobe Systems Inc. | http://www.graphpad.com |
| Software, algorithm | GeneChip Scanner 3000 7G | Affymetrix | N/A |
| Software, algorithm | Expression Console | Affymetrix | https://imagej.nih.gov/ij/ |
| Software, algorithm | DIVA8.0.1 | BD Biosciences | N/A |
| Software, algorithm | R | Bioconductor | N/A |
| Software, algorithm | Prism 7 | GraphPad | N/A |
| Software, algorithm | ImageJ | NIH | https://www.bioconductor.org/ |
| Software, algorithm | Partek Genomics Suite | Partek | N/A |
| Deposited data | GSE14308 | *Wei et al. (2009)*, PMID: 19144320 | https://www.ncbi.nlm.nih.gov/geo/query/acc.cgi?acc=GSE14308 |

*Continued on next page*

*Continued*

| Reagent type (species) or resource | Designation | Source or reference | Identifiers |
|---|---|---|---|
| Deposited data | GSE42276 | *Wakamatsu et al. (2013)*, PMID: 23277554 | https://www.ncbi.nlm.nih.gov/geo/query/acc.cgi?acc=GSE42276 |
| Deposited data | GSE67464 | *Bevington et al., 2016*, PMID: 26796577 | https://www.ncbi.nlm.nih.gov/geo/query/acc.cgi?acc=GSE67464 |
| Deposited data | GSE70154 | *Richards et al. (2015)*, PMID: 26195815 | https://www.ncbi.nlm.nih.gov/geo/query/acc.cgi?acc=GSE70154 |
| Deposited data | GSE62532 | *Vahl et al. (2014)*, PMID: 25464853 | https://www.ncbi.nlm.nih.gov/geo/query/acc.cgi?acc=GSE62532 |
| Deposited data | GSE97477 | This paper | https://www.ncbi.nlm.nih.gov/geo/query/acc.cgi?acc=GSE97477 |

## Mice

C57BL/6 mice (CD45.2) were obtained from Charles River Laboratories. C57BL/6 CD45.1 mice were maintained in our own animal facilities, under specific pathogen-free conditions. C57BL/6 Foxp3-GFP CD45.2 mice (*Wang et al., 2008*), initially obtained from Dr Bernard Malissen, Centre d'Immunologie de Marseille-Luminy, France, were crossed with C57BL/6 CD45.1 mice to generate C57BL/6 Foxp3-GFP CD45.1 and CD45.1/.2 mice. C57BL/6 OT-II mice were obtained from Charles River Laboratories and crossed with C57BL/6 Foxp3-GFP CD45.1 (or CD45.2) mice to generate C57BL/6 Foxp3-GFP CD45.1/.2 (or CD45.2) OT-II mice. Four- to 12-week-old mice were used for all experiments. Experiments were carried out in accordance with the guidelines of the French Veterinary Department. All procedures performed were approved by the Paris-Descartes Ethical Committee for Animal Experimentation (decision CEEA34.CA.080.12). Sample sizes were chosen to ensure the reproducibility of the experiments and according to the 3Rs of animal ethics regulation.

## Cell suspensions

Peripheral Lymph Nodes (pLNs), mesenteric Lymph Nodes (mLNs), Peyer's patches, spleen and thymus were homogenized and passed through a nylon cell strainer (BD Falcon) in PBS supplemented with 10% FCS (Biochrom) for adoptive transfer or cell culture (LNs only), or in 5% FCS and 0.1% $NaN_3$ (Merck-Sigma-Aldrich, Lyon, France) in PBS for flow cytometry.

## Adoptive transfer of Ly-6C$^-$ CD4 $T_N$ cells

CD4 T cells were purified from LNs (pooled superficial cervical, axillary, brachial, inguinal and mLNs) of C57BL/6 Foxp3-GFP CD45.1 mice by incubating cell suspensions on ice for 15 min with a mixture of anti-CD8 (53–6.7), anti-CD19 (1D3) and anti-Ter-119 antibodies (Abs) obtained from hybridoma supernatants, and then with magnetic beads coupled to anti-rat immunoglobulins (Invitrogen, Cergy-Pontoise, France). Ly-6C$^-$ CD4 $T_N$ cells were sorted as Foxp3-GFP$^-$ Lineage (CD25, TCRγδ, CD8β, CD11b, CD11c)-PE$^-$ CD44$^{-/lo}$ Ly-6C$^-$ cells using a FACS-ARIA3 flow cytometer (BD Biosciences, Le Pont de Claix, France) and injected intravenously into sex-matched recipient mice whose then were injected intraperitoneally every day for two weeks with 2.5 mg/kg of Prograf (Tacrolimus; Astellas Pharma Inc., Tokyo, Japan).

## Adoptive transfer of OT-II CD4 $T_N$ cells

CD4 T cells were purified from LNs of C57BL/6 Foxp3-GFP OT-II CD45.2 or CD45.1/.2 mice by using Dynabeads Untouched Mouse CD4 Cells Kit (Invitrogen) and cultivated with recombinant mouse IL-7 (10 ng/ml; R and D Systems, Minneapolis, MN) with or without Thapsigargin (4 nM; Merck-Sigma-Aldrich) into 96-well round-bottom treated cell culture microplate (Corning; $1 \times 10^5$ cells per well). After 5 days of culture, cells were recovered and labelled with PE-conjugated anti-TCRγδ (GL3), anti-CD8.b2 (53–5.8), anti-NK-1.1 (PK136) and APC-conjugated anti-CD44 (IM7), all from BD Biosciences. OT-II CD4 $T_N$ cells were sorted as GFP$^-$ Lineage-PE$^-$ CD44$^{-/lo}$ cells using a FACS-ARIA3 flow cytometer (BD Biosciences) and 0.5 to $1 \times 10^6$ cells were injected intravenously into sex-matched C57BL/6 Foxp3-GFP CD45.1 mice. Recipient mice were then continuously fed with Albumin from chicken egg white (OVA; 1.5% w/v; Merck-Sigma-Aldrich) in the drinking water or not. LNs and spleens were

collected at day seven and CD45.2$^+$ CD4 T cells analyzed. In a second protocol, sorted CD4 T$_N$ cells from CD45. 2$^+$ and CD45.1/.2$^+$ C57BL/6 Foxp3-GFP OT-II mice were cultured in IL-7 (10 ng/ml) without or with TG (4 nM), respectively. After 5 days live CD4 T$_N$ (CD44$^{lo}$ CD25$^{lo}$ CD8$\beta^-$ CD11b$^-$ CD11c$^-$ NK1.1$^-$ TCR$\gamma\delta^-$ Foxp3-GFP$^-$) cells were flow-cytometry sorted, mixed at a 1:1 ratio and injected intravenously (0.5−1 × 10$^6$ cells) into sex-matched CD45.1$^+$ C57BL/6 Foxp3-GFP recipient mice gavaged with Ovalbumin (OVA; 50 mg) 4 and 24 hr later. LNs and spleens were collected at day 10 and donor-derived CD4 T cells were analyzed.

## Cell surface staining and flow cytometry

Cell suspensions were collected and dispensed into 96-well round-bottom microtiter plates (Greiner Bioscience; 6 × 10$^6$ cells per well). Surface staining was performed by incubating the cells on ice, for 15 min per step, with Abs in 5% FCS and 0.1% NaN$_3$ in PBS. Each cell-staining reaction was preceded by a 15 min incubation with a purified anti-mouse CD16/32 Abs (Fc$\gamma$RII/III block; 2.4G2) obtained from hybridoma supernatants.

Alexa Fluor 700-conjugated anti CD45.2 (104), Allophycocyanin (APC)-conjugated anti-CD25 (PC61) and anti-CD44 (IM7), Brilliant Violet (BV) 421-conjugated anti Ly-6C (AL-21), BV 510-conjugated anti-CD4 (RM4-5), BV 786-conjugated anti-CD25 (PC61), Phycoerythrin (PE)-conjugated anti-CD25 (PC61), anti-CD69 (H1.2F3), anti-Izumo1r (TH6), anti-TCR$\gamma\delta$ (GL3) and anti-V$\beta$5.1/5.2 (MR9-4), PE-Cy7-conjugated anti-CD44 (IM7) and anti-CD45.1 (A20), biotinylated anti-CD5 (53–7.3), anti-CD62L (MEL14), anti-Ly-6C (AL-21) and anti-Sca1 (E13-161.7) were obtained from BD Biosciences. Alexa Fluor 647-conjugated anti-IL18r$\alpha$ (BG/IL18r$\alpha$), APC-conjugated streptavidin, BV 421-conjugated anti-Ly-6C (HK1.4) and PE-conjugated anti-Ly-6C (HK1.4) were obtained from BioLegend (London, United Kingdom). PE-conjugated anti-CD200 (OX-90), anti-Ikzf3 (8B2) and anti-Nur77 (12.14), PerCP-Cy5.5-conjugated anti-TCR$\beta$ (H57-597) and biotinylated anti-CD73 (eBioTY/11.8) were obtained from eBioscience (Montrouge, France). Pacific Blue-conjugated streptavidin was obtained from Invitrogen. APC-Vio770-conjugated anti-CD8$\alpha$ (53–6.7) and PE-conjugated anti-CD122 (TM-$\beta$1) were obtained from Miltenyi Biotec.

Multi-colour immunofluorescence was analyzed using BD-LSR2 and BD-FORTESSA (BD Biosciences) flow-cytometers. List-mode data files were analyzed using Diva software (BD Biosciences). Data acquisition and cell sorting were performed on the Cochin Immunobiology facility.

## Intracellular calcium measurement

Ex vivo purified CD4 T cells or cells recovered after 5 days of culture were loaded for 30 min at 37°C with the membrane-permeable fluorescent Ca$^{2+}$ indicator dye Indo-1 AM (Invitrogen) at a concentration of 1 μM. Cells were stained either in HBSS (for ex-vivo-purified CD4 T cells) or directly in the culture medium (cultured cells). Thereafter, ex-vivo-purified CD4 T cells were stained for surface markers and kept on ice. Before acquisition, cell aliquots were allowed to equilibrate to 37°C for 5 min and then were analyzed by flow cytometry. After acquisition of background intracellular Ca$^{2+}$ concentrations for 2 min, cells were stimulated with Thapsigargin (at a concentration of 4 or 200 nM).

## Cell culture and in vitro polarization assays

Flow-cytometry sorted Ly-6C$^-$ and Ly-6C$^+$ CD4 T$_N$ cells from LNs of C57BL/6 Foxp3-GFP mice were stained with CellTrace Violet (CTv; 5 μM; Life Technologies) and cultured with IL-7 (10 ng/ml) alone or in combination with Thapsigargin (TG; 4 nM), Phorbol 12-myristate 13-acetate (PMA; 1.25 ng/ml), PMA +TG (1.25 ng/ml and 4 nM, respectively) and immobilized anti-CD3 (clone 145.2C11; 4 μg/ml; obtained from hybridoma supernatants) and anti-CD28 (clone 37.51; eBioscience; 4 μg/ml) Abs. For in vitro polarization assays Ly-6C$^+$ CD4 T$_N$ cells were additionally stained with CellTrace Far Red (CTfr; 1.25 μM; Life Technologies). Cells were then stimulated separately or together for 4 days with coated anti-CD3 and anti-CD28 Abs, in the presence of graded concentrations of exogenous recombinant human TGF$\beta$1 (Invitrogen). In some experiments, anti-IFN-$\gamma$ (clone R4-6A2; 10 μg/mL) and anti-IL-4 (clone 11B11; 10 μg/mL) blocking antibodies were added in the culture.

The concentration of TGF$\beta$ needed to obtain 50% of the maximal percentage of iTreg cells (Effective Concentration, EC50) was calculated by fitting the dose-response curves of CD4 T$_N$-cell subsets in the different culture conditions. To this end, the means of 3 to 5 independent experiments were

used to build dose response curves using nonlinear least-squares regression to the Hill equation. The model used for this function was $Y=[B+(T-B)] / [1 + 10^{([LogEC50-X]*HillSlope)}]$, where 'Y' represents Foxp3$^+$ cells as a percentage among CD4$^+$ cells, 'T' and 'B' represent the plateaus at the beginning and end of the curve, respectively, and 'X' represents the concentration of TGFβ added at the beginning of the culture. The absolute EC50 was calculated to interpolate X at 50% with 95% confidence intervals.

## Cytokine multiplex assay

Flow-cytometry sorted Ly-6C$^-$ and Ly-6C$^+$ CD4 T$_N$ cells from LNs of C57BL/6 Foxp3-GFP mice were stimulated as described above with immobilized anti-CD3 and anti-CD28 Abs in the presence or absence of exogenous recombinant human TGFβ1 (Invitrogen, 4 µg/mL). Supernatants were recovered 24 hr later and cytokines were quantified by MSD multi-array U-PLEX assays (IFN-γ, IL-4, IL-17A/F and IL-10; Meso Scale Discovery, Rockville, MD) according to the manufacturer's instructions.

## Imaging flow cytometry

LNs cells of C57BL/6 mice were harvested and fixed in 4% paraformaldehyde, immediately or after 30 min of resting or stimulation with 200 nM of Thapsigargin in RPMI 1640 Glutamax (Gibco). Cells were washed in 1% FCS and 0.1% NaN$_3$ in PBS and incubated in glycine (0.1M) for 10 min. Cell surface was stained with biotinylated anti-Ly-6C (AL-21), BV 510-conjugated anti-CD4 (RM4-5), PE-conjugated anti-CD25 (PC61), anti-TCRγδ (GL3), anti-CD8.β2 (53–5.8), anti-NK-1.1 (PK136), anti-CD11b (M1/70), PE-Cy7-conjugated anti-CD44 (IM7) and PerCp-Cy5.5-conjugated streptavidin, all from BD Biosciences. Intracellular stainings were performed using Foxp3 Staining kit (eBioscience) and Alexa 448-conjugated anti-NFAT1 (D43B1; Cell Signaling, Leiden, The Netherlands) or anti-NFAT2 (7A6; BioLegend) and APC-conjugated anti-Foxp3 (FJK-165; eBioscience) Abs were used. Ly-6C$^-$ and Ly-6C$^+$ CD4 T$_N$ cells were sorted as CD4-BV510$^+$ Lineage-PE$^-$ CD44$^{-/lo}$ Foxp3-APC$^-$ Ly-6C$^{+/-}$ cells using a FACS-ARIA3 flow cytometer (BD Biosciences). After sort, DRAQ5 (Cell Signaling) was used to stain nuclei. Cells were acquired with ImageStreamX (Amnis; EMD Millipore) and analyzed with IDEAS software. NFAT1 and NFAT2 nuclear localization was calculated as the similarity score between NFAT and DRAQ5 intensities.

## Microarray

CD4 T cells from LNs of C57BL/6 Foxp3-GFP mice were enriched as described above. Then, Ly-6C$^-$ and Ly-6C$^+$ CD4 T$_N$ cells were flow-cytometry sorted as CD4$^+$ CD8α$^-$ TCRβ$^+$ GFP$^-$ CD25$^-$ CD44$^{-/lo}$ cells using a FACS-ARIA3 flow cytometer. Total RNA was extracted using the RNeasy Mini kit (QIAGEN, Courtaboeuf, France). RNA quality was validated with Bioanalyzer 2100 (using Agilent RNA6000 nano chip kit). Experimental and analytical part of the microarray analysis was performed according to the MIAME standards. Amplified, fragmented and biotinylated sense-strand DNA targets were synthesized from 50 ng total RNA according to the manufacturer's protocol (Ovation PicoSL WTA System V2 and Encore Biotin Module kit (Nugen, Leek, The Netherlands)) and hybridized to a mouse gene 2.0 ST array (Affymetrix, Paris, France). The stained chips were read and analysed with a GeneChip Scanner 3000 7G and Expression Console software (Affymetrix). Raw data (.cel files) were then processed and normalized using the quantile normalization method in RMA with R package (Bioconductor). Statistical analysis was then performed with Partek Genomics Suite software (Partek). Gene expression was z-transformed, for visualization, using the following formula: $z=(X-\mu)/\tau$, with X = normalized intensity, μ = mean of the normalized intensity across replicates and τ = s.d. of mean of the normalized intensity across replicates. Experimental and analytical part of the microarray was performed on the Cochin Genomic facility. Raw and processed data microarray data are provided in the Gene Expression Omnibus (GEO) under accession number GSE97477.

## Comparison with public GEO datasets

Normalized microarray datasets (GSE14308, GSE42276, GSE67464, GSE70154 and GSE62532) were recovered from NCBIs Gene Expression Omnibus (GEO, http://www.ncbi.nlm.nih.gov/geo/). For each datasets the values mean of Probset with the same Gene-id was performed to generate a file (.xlsx) with a unique value per Gene-id for each sample. These files were then statistical analyzed as described above. The newly created public GEO Datasets were then aligned with our microarray

data by keeping only the commons Gene-id. Finally, these alignment files were filtered on our data for a p-value<0.05 and a fold change >1.3 and the differential expression of genes was compared between our and public GEO microarray.

## Acknowledgements

We greatly acknowledge E Maillard, K Labroquère and M Andrieu from the Cochin Cytometry and Immunobiology (CYBIO) facility, S Jacques and F Letourneur from the Cochin Genomic (GENOM'IC) facility and P Bourdoncle and B Durel from the Cochin Imaging (IMAG'IC) facility. This work was supported by grants from the 'Fondation pour la Recherche Médicale' (FRM team), the 'Ligue contre le Cancer', the 'Association pour la Recherche sur le Cancer' and the 'Agence nationale de la recherche' (grant number ANR-15-CE15-0009-01). V Guichard was supported by a PhD fellowship from the French Ministry of National Education, Research, and Technology. A Audemard-Verger was supported by a PhD fellowship from the 'Ligue contre le Cancer' and from INSERM. We would like to thank B Salomon, A Boissonnas, C Combadière, C Randriamanpita, A Trautmann, G Bismuth, E Donnadieu and G Eberl for providing tools and/or advices.

## Additional information

### Funding

| Funder | Grant reference number | Author |
|---|---|---|
| Institut National de la Santé et de la Recherche Médicale | | Nelly Bonilla<br>Alexandra Audemard-Verger<br>Thomas Guilbert<br>Bruno Martin<br>Bruno Lucas<br>Cédric Auffray |
| Agence Nationale de la Recherche | ANR-15-CE15-0009-01 | Cédric Auffray |
| Fondation pour la Recherche Médicale | | Bruno Lucas |
| Ligue Contre le Cancer | | Alexandra Audemard-Verger |

The funders had no role in study design, data collection and interpretation, or the decision to submit the work for publication.

### Author contributions

Vincent Guichard, Conceptualization, Formal analysis, Investigation, Writing—original draft, Writing—review and editing; Nelly Bonilla, Aurélie Durand, Alexandra Audemard-Verger, Investigation, Methodology; Thomas Guilbert, Software, Investigation; Bruno Martin, Funding acquisition, Validation, Investigation; Bruno Lucas, Conceptualization, Data curation, Supervision, Funding acquisition, Project administration, Writing—review and editing; Cédric Auffray, Conceptualization, Data curation, Formal analysis, Supervision, Funding acquisition, Validation, Investigation, Methodology, Writing—original draft, Project administration, Writing—review and editing

### Author ORCIDs

Vincent Guichard (iD) http://orcid.org/0000-0003-0206-4924
Cédric Auffray (iD) http://orcid.org/0000-0002-1012-0132

### Ethics

Animal experimentation: Experiments were carried out in accordance with the guidelines of the French Veterinary Department. All procedures performed were approved by the Paris-Descartes Ethical Committee for Animal Experimentation (decision CEEA34.CA.080.12). Sample sizes were chosen to ensure the reproducibility of the experiments and according to the 3Rs of animal ethics regulation.

Decision letter and Author response
Decision letter https://doi.org/10.7554/eLife.27215.031
Author response https://doi.org/10.7554/eLife.27215.032

## Additional files

### Supplementary files
• Transparent reporting form
DOI: https://doi.org/10.7554/eLife.27215.017

### Major datasets
The following dataset was generated:

| Author(s) | Year | Dataset title | Dataset URL | Database, license, and accessibility information |
|---|---|---|---|---|
| Guichard V, Bonilla N, Durand A, Audemard-Verger A, Guilbert T, Martin B, Lucas B, Auffray C | 2017 | Calcium-mediated shaping of naive CD4 T cell phenotype and function | https://www.ncbi.nlm.nih.gov/geo/query/acc.cgi?acc=GSE97477 | Publicly available at the NCBI Gene Expression Omnibus (accession no. GSE97477) |

The following previously published datasets were used:

| Author(s) | Year | Dataset title | Dataset URL | Database, license, and accessibility information |
|---|---|---|---|---|
| Wei L, Wei G, Zhu J, Hu-Li J, O'Shea JJ, Zhao K | 2009 | Epigenetic Mechanisms Underlie T Cell Plasticity | https://www.ncbi.nlm.nih.gov/geo/query/acc.cgi?acc=GSE14308 | Publicly available at the NCBI Gene Expression Omnibus (accession no. GSE14308) |
| Wakamatsu E, Mathis D, Benoist C | 2012 | Gene expression profile of conventional T cells (Tconv) and regulatory T cells (Treg) stimulated with anti-costimulatory molecule antibodies | https://www.ncbi.nlm.nih.gov/geo/query/acc.cgi?acc=GSE42276 | Publicly available at the NCBI Gene Expression Omnibus (accession no. GSE42276) |
| Bevington S, Cauchy P, Jason P, Elizabeth B, Naveen L, Ott S, Bonifer C, Cockerill P | 2016 | Defining the molecular mechanisms underlying immunological memory in T cells (expression) | https://www.ncbi.nlm.nih.gov/geo/query/acc.cgi?acc=GSE67464 | Publicly available at the NCBI Gene Expression Omnibus (accession no. GSE67464) |
| Richards DM, Hofer A, Feuerer M | 2015 | The contained self-reactive peripheral T cell repertoire: size, diversity and cellular composition | https://www.ncbi.nlm.nih.gov/geo/query/acc.cgi?acc=GSE70154 | Publicly available at the NCBI Gene Expression Omnibus (accession no.GSE70154) |
| Vahl JC, Schallenberg S, Buch T, Kretschmer K, Schmidt-Supprian M | 2014 | Continuous T cell receptor signals maintain a functional regulatory T cell pool | https://www.ncbi.nlm.nih.gov/geo/query/acc.cgi?acc=GSE62532 | GSE62532 |

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
