## [Decision Letter]

Thank you for submitting your article "Calcium-mediated shaping of naive CD4 T cell phenotype and function" for consideration by *eLife*. Your article has been reviewed by three peer reviewers, one of whom, Michael L Dustin (Reviewer #1), is a member of our Board of Reviewing Editors, and the evaluation has been overseen by Tadatsugu Taniguchi as the Senior Editor.

The reviewers and editors feel that your paper (Auffray et al) potentially complements an independently submitted paper. You were contacted about this opportunity and agreed to exchange information with the authors of that paper after receiving the three individual reviews for each. The following is a consensus review among three of the reviewers who agreed to oversee a co-revision. The intention is to help you prioritise the revision and also to clarify the opportunity for coordination from our perspective.

The two papers deal with different consequences of calcium mobilization in vivo. Auffray et al., focuses on transcriptional changes that are brought about in the most self reactive naïve T cells in mice that lead to loss of Ly6C expression and a tendency to differentiate into peripheral Treg upon activation. This state could be mimicked by treatments that may chronically enhance cytoplasmic calcium, but the evidence for any effects on cytoplasmic calcium ion levels by self pMHC in vivo or low concentrations of thapsigargin in vitro are lacking, which is a major gap. There are also mechanistic aspects related to cytokine production in the Ly6C positive population that could be explored.

Cahalan et al. focuses on the role of CRAC channel activation in T cell trafficking and interstitial migration in the T cell zone of lymph nodes. A unique aspect of the paper is that they focus entirely on human T cells in profoundly immunodeficient mice. CRAC is eliminated through expression of a dominant negative Orai, which is well established to eliminate sustained calcium increases. The authors developed a new calcium reporter based on fusing GCaMP6f to tdTomato to make a ratiometric reporter called Salsa6f, which is sufficient sensitive to detect calcium spikes in intact lymph nodes. The main points are that CRAC channel function is needed for efficient lymph node entry and that cell-autonomous calcium fluxes result in slowing of T cell migration. The weakness in this paper is that the actual mechanism of the calcium fluxes is not established and objective statistical tests supporting a relationship are not provided. The trigger for the calcium fluxes observed in not clear.

The editors felt that there is strong possibility that addressing issues in Cahalan et al., may provide key supporting data for Auffray.et al. And reciprocally, data in Auffray et al. provides further physiological significance for data in Cahalan.et al. The authors may therefore benefit from coordinating on the revisions and the authors have agreed to exchange information. The editors have prepared the following joint consensus review following provision of the individual reviews.

1) The main concern with Auffray et al. is that there is no data on the impact of pharmacological manipulations on cytoplasmic calcium levels in the naïve mouse T cells. The authors of Cahalan et al. are experts in this area. It will be important for the authors to use in vitro calcium studies to determine the impact of chronic exposure to 4 nM thapsigargin in different time windows- does this result in sustained cytoplasmic calcium elevation capable to inducing NFAT translocation. This may require a highly sensitive method and advice from authors of Cahalan et al. should be helpful to address this concern in a 2 month time frame. But this concern needs to be addressed.

2) The main concern with Cahalan et al. is that the mechanism triggering the calcium fluxes in vitro and in vivo is not clear. T cell migrating on flat substrates didn't display the calcium fluxes, but movement in PDMS channels and within lymph nodes did. The authors attention was called to recent studies that suggest that surface interactions could trigger calcium signals related to formation of close contacts that exclude CD45. Thus, some cellular confinement in the PDMS channels may trigger signals. The in vivo situation is of greatest interest. The use of the human-mouse system is a complication as its not entirely clear self pMHC is supplied by human monocyte derived DC or mouse xeno-MHC, which can trigger T cell responses. In murine systems, particularly on H2-b haplotype, it is relatively easy to eliminate self pMHC signals in CD4^+^ T cells acutely by blocking I-Ab with an antibody (e.g. Stefanova I, Dorfman JR, Germain RN. Self-recognition promotes the foreign antigen sensitivity of naive T lymphocytes. Nature. 2002;420(6914):429-34.) or by transferring CD4 T cells in to I-Ab deficient mice. Thus, while the human mouse system has some utility as a preclinical model, it's a relatively weak tool to study basic mechanisms in this case as even if only human pMHC is recognized, there are multiple possible class I and class II genes that would need to be blocked to eliminate all possible recognition. The author should repeat a key experiment in a mouse model system where a role of self-pMHC in the in vivo calcium fluxes could be supported or ruled out. Introduction of the Salsa6f reporter into naïve T cells would be helpful in this context. If the authors happen to have generated a transgenic model with Salsa6f or have the ability to introduce it by RNA electroporation then this could be combined with antibody blocking to test a role for self-pMHC in the calcium activity independent of the DN Orai construct, which is not necessarily needed to ask this question. The result of this experiment would also provide key information for Auffray.et al.

Additional points:

1) Are the Ly6C negative naïve T cell poor producers of cytokines that would prevent Treg induction in vitro? This could be tested with a number of systems to evaluate cytokine production.

2) The oral tolerance studies should be performed at different antigen doses to be convincing.

---

## [Author Response]

1) The main concern with Auffray et al. is that there is no data on the impact of pharmacological manipulations on cytoplasmic calcium levels in the naïve mouse T cells. The authors of Cahalan et al. are experts in this area. It will be important for the authors to use in vitro calcium studies to determine the impact of chronic exposure to 4 nM thapsigargin in different time windows- does this result in sustained cytoplasmic calcium elevation capable to inducing NFAT translocation. This may require a highly sensitive method and advice from authors of Cahalan et al. should be helpful to address this concern in a 2 month time frame. But this concern needs to be addressed.

We have now performed experiments to determine the impact of chronic exposure to 4 nM thapsigargin. Calcium mobilization induced by 4nM thapsigargin has been compared to the one induced by the classical dose of 200nM. In the revised version of our manuscript, we are showing that 4nM TG induce a moderate but significant increase in intracellular calcium levels, although to a lesser extent than the 200nM dose. We have added these data in Figure 4 (panel C) and described them in the Results section of our manuscript:

“Importantly, to avoid TG induced cell death, a sub-optimal dose (4nM) was used in these culture conditions. 4nM TG induced a reproducible increase in intracellular calcium levels, although to a lesser extent than the classical dose of 200nM (Figure 4).”

To better characterize the long-term effect of 4nM TG during the culturing period, baseline calcium cytoplasmic levels were assessed after 5 days of culture. Cells cultured in IL-7 alone were used as control. We observed that, even at this late time point, 4nM TG treated Ly-6C^+^ CD4 T_N_ cells still exhibited higher cytoplasmic Ca^2+^ levels than control cells. To strengthen this result, and as requested by the reviewers, we have also analyzed NFAT subcellular localization at various time points throughout the culture. NFAT was translocated into the nucleus of 4nM thapsigargin treated cells as soon as 1 day after the beginning of the culture. More precisely, NFAT translocation into the nucleus peaked on day 1 and remained significantly higher in TG treated cells than in control cells throughout the culture. We have now added these data in Figure 4 (panels D-F) and described them in the Results section of our manuscript:

“After 5 days of culture, 4nM TG treated Ly-6C^+^ CD4 T_N_ cells still exhibited higher cytoplasmic Ca^2+^ levels than control cells cultured in IL-7 alone (Figure 4). To further characterize the long-term effect of this low dose TG, subcellular localization of the nuclear factor of activated T cell protein 1 (NFAT1) was assessed in Ly-6C^+^ CD4 T_N_ cells in the presence or absence of 4nM TG at various time points along the culture. Indeed, increases in intracellular Ca^2+^ levels result in the activation of Calcineurin that dephosphorylates members of the NFAT family, leading to their translocation into the nucleus. NFAT1 localization was quantified by high-resolution imaging flow-cytometry using the ImageStreamX technology (Figure 4). In line with the Ca^2+^ increase induced by 4nM TG treatment, NFAT was translocated into the nucleus of Ly-6C^+^ CD4 T_N_ cells in the presence of TG while it remained cytoplasmic in its absence. NFAT translocation into the nucleus peaked on day 1 and remained significantly higher in TG treated cells than in control cells throughout the culture.”

Furthermore, our data demonstrating that the in vivo inhibition of Calcineurin activity led to the conversion of Ly-6C^-^ to Ly-6C^+^ CD4 T_N_ cells were reinforced by imaging flow cytometry data showing that such a treatment also led to the rapid exclusion of NFAT from the nucleus of Ly-6C^-^ CD4 T_N_ cells. Of note, a short-term exposure (18 hours) to FK506 was used in this experiment to prevent the phenotypic conversion of Ly-6C^-^ to Ly-6C^+^ CD4 T_N_ cells. We have added these data in Figure 5—figure supplement 1 and described them in the Results section of our manuscript:

“We first confirmed that the Ca^2+^-Calcineurin signaling cascade was active in vivo in Ly-6C^-^ CD4 T_N_ cells by showing that blocking Calcineurin activation for 18 hours with FK506 was sufficient to abrogate the nuclear localization of NFAT in these cells (Figure 5—figure supplement 1). We then wondered whether a longer treatment with this Calcineurin inhibitor would affect the phenotype of CD4 T_N_ cells in vivo.”

Additional points:1) Are the Ly6C negative naïve T cell poor producers of cytokines that would prevent Treg induction in vitro? This could be tested with a number of systems to evaluate cytokine production.

Such an hypothesis is consistent with the literature showing that the high levels of CD5 expressed by CD5^high^ CD4 T_N_ cells counteract the inhibitory effect of T_H_1- and T_H_2-cell derived cytokines on iTreg-cell induction (J.G. Henderson et al., Immunity 2015, PMID: 25786177). As Ly-6C^-^ CD4 T_N_ cells exhibit higher levels of CD5 than Ly-6C^+^ CD4 T_N_ cells, it could have been thus consistent with our data showing that when co-cultured with Ly-6C^+^ CD4 T_N_ cells, Ly-6C^-^ CD4 T_N_ cells still differentiated more efficiently than their Ly-6C^+^-cell counterparts into iTreg cells in vitro.

We are now showing that Ly-6C^-^ and Ly-6C^+^ CD4 T_N_ cells, stimulated for 24 hours with αCD3- and αCD28-coated antibodies in the presence or absence of TGFβ, produce similar amounts of IFN-γ, IL-4, IL-10 and IL-17 (Figure 1—figure supplement 1). We have added these data in a new figure supplement (Figure 1—figure supplement 1) and described them in the Results and Discussion sections of our manuscript:

“As T_H_1- and T_H_2-cell derived cytokines are known to inhibit iTreg-cell induction in vitro (références), we first wondered whether Ly-6C^-^ and Ly-6C^+^ CD4 T_N_ cells had the same ability to produce such cytokines after stimulation. Ly-6C^-^ and Ly-6C^+^ CD4 T_N_ cells were thus stimulated with αCD3- and αCD28-coated antibodies in the presence or absence of TGFβ. Interferon-γ (IFN-γ) and interleukins (IL) -4, -17 and -10 were assayed in the supernatants collected 24 hours after the beginning of the culture. We found that, whatever the presence or absence of TGFβ in the culture medium, Ly-6C^-^ and Ly-6C^+^ CD4 T_N_ cells produced similar amounts of these cytokines (Figure 1—figure supplement 1).”

“Indeed, Ly-6C^-^ and Ly-6C^+^ CD4 T_N_ cells produced similar amounts of these cytokines after stimulation and Ly-6C^-^ CD4 T_N_ cells still differentiated more efficiently than their Ly-6C^+^-cell counterparts into iTreg cells in vitro in the presence of anti-cytokine (IL-4 and IFN-γ) blocking antibodies (Figure 1—figure supplement 1).”

Moreover, we have supplemented these data with an in vitro polarization experiment comparing the ability of Ly-6C^-^ and Ly-6C^+^ CD4 T_N_ cells to convert into iTreg cells in the presence of anti-IFN-γ and anti-IL-4 blocking mAbs. In line with our cytokine assay, adding these blocking mAbs in the culture medium does not abolish the difference in the ability of Ly-6C^-^ and Ly-6C^+^ CD4 T_N_ cells to differentiate into iTreg cells. The result of this experiment was previously quoted in the Discussion section, as data not shown, and is now included both in the Results and the Discussion sections of our revised manuscript (Figure 1—figure supplement 1 and in the text):

“Of note, and in line with their similar ability to produce T_H_1- and T_H_2-cell derived cytokines, blocking IFN-γ and IL-4 during in vitro iTreg-cell polarization did not abolish the difference in the ability of Ly-6C^-^ and Ly-6C^+^ CD4 T_N_ cells to differentiate into iTreg cells. (Figure 1—figure supplement 1).”

2) The oral tolerance studies should be performed at different antigen doses to be convincing.

First, we have corrected an error in the original Figure 7. Indeed, in this model, OVA was continuously given in the drinking water (1.5% (w/v)). This point was already specified in the Material and methods section as well as in the figure legend of the first version of our manuscript. To strengthen this part of our results, instead of testing different antigen doses in our initial protocol (Such an experiment would have of course considerably strengthen our results but would have required important numbers of C57BL/6 Foxp3-GFP OT-II mice), we have chosen to use a second well-defined protocol of oral administration of OVA. More precisely, in this setting, 50 mg OVA were given by gavage (4 and 24 hours after the transfer of OT-II cells). Moreover, in this setting, “Ca^2+^-converted” and control OT-II CD4 T_N_ cells were co-transferred in order to compare their ability to convert into pTreg cells in the same recipient mice. 10 days after transfer (instead of 7 in the first mouse model), we demonstrated that “Ca^2+^-converted” OT-II CD4 T_N_ cells gave rise to greater absolute numbers of Foxp3-expressing cells than control OT-II CD4 T_N_ cells. We have added these data in a new figure supplement (Figure 7—figure supplement 1) and described them in the Results section of our manuscript:

“These latter results were confirmed by using a second protocol of oral administration of OVA. In this setting, “Ca^2+^-converted” OT-II CD4 T_N_ cells were co-transferred with OT-II CD4 T_N_ cells cultured in IL-7 alone in order to compare their ability to convert into pTreg in the same recipient mice. FACS-sorted CD45. 2^+^ and CD45.1/2^+^ OT-II CD4 T_N_ cells were first cultured in IL-7 in the absence or presence of TG respectively (Figure 7—figure supplement 1). After 5 days of culture, living cells were FACS-sorted, mixed at a 1:1 ratio and 1.10^6^ cells were adoptively transferred into CD45.1 Foxp3-GFP mice. Finally, recipient mice were fed with OVA by gavage (4 and 24 hours after the transfer of OT-II cells). Nine days later, secondary lymphoid organs were recovered and the phenotype of donor-derived OT-II T cells was analyzed. In this setting, “Ca^2+^-converted” OT-II CD4 T_N_ cells were also giving rise to greater absolute numbers of Foxp3-expressing cells than OT-II CD4 T_N_ cells cultured with IL-7 alone prior to injection (Figure 7—figure supplement 1). More precisely, more than three quarters of the Foxp3^+^ OT-II cells arising in these conditions derived from “Ca^2+^-converted” cells (Figure 7—figure supplement 1).”